# MLL-AF9 initiates transformation from fast-proliferating myeloid progenitors

Xinyue Chen [1,2], Daniel B. Burkhardt [3], Amaleah A. Hartman[1,2], Xiao Hu[1,2], Anna E. Eastman[1,2], Chao Sun[1,2], Xujun Wang [4], Mei Zhong[1,2], Smita Krishnaswamy [3]* & Shangqin Guo [1,2]*

Cancer is a hyper-proliferative disease. Whether the proliferative state originates from the cell-of-origin or emerges later remains difficult to resolve. By tracking de novo transformation from normal hematopoietic progenitors expressing an acute myeloid leukemia (AML) oncogene MLL-AF9, we reveal that the cell cycle rate heterogeneity among granulocyte–macrophage progenitors (GMPs) determines their probability of transformation. A fast cell cycle intrinsic to these progenitors provide permissiveness for transformation, with the fastest cycling 3% GMPs acquiring malignancy with near certainty. Molecularly, we propose that MLL-AF9 preserves gene expression of the cellular states in which it is expressed. As such, when expressed in the naturally-existing, rapidly-cycling immature myeloid progenitors, this cell state becomes perpetuated, yielding malignancy. In humans, high *CCND1* expression predicts worse prognosis for MLL fusion AMLs. Our work elucidates one of the earliest steps toward malignancy and suggests that modifying the cycling state of the cell-of-origin could be a preventative approach against malignancy.

[1] Department of Cell Biology, Yale University, New Haven, CT 06520, USA. [2] Yale Stem Cell Center, Yale University, New Haven, CT 06520, USA. [3] Department of Genetics, Yale University, New Haven, CT 06520, USA. [4] SJTU-Yale Joint Center for Biostatistics and Data Science, Shanghai Jiao Tong University, Shanghai 200240, China. *email: smita.krishnaswamy@yale.edu; shangqin.guo@yale.edu

Not all cells carrying oncogenes result in malignant transformation[1], raising the question of what discriminates a cell to be transformed from those to remain normal despite their shared genetic abnormalities. A frequently considered scenario is the acquisition of additional genetic lesions, allowing mutant cells to progressively gain proliferative advantage, evade apoptosis, and/or immune surveillance[2], leading to their net expansion. While this multihit oncogenic model has extensive support from solid cancers[3–5], several types of malignancy have rather low mutational load, such as those of hematopoietic origin[5,6]. Expression of a single oncogene, such as the mixed lineage leukemia (MLL) fusion oncogenes, is often sufficient to induce malignancy in animal models[7–9]. Alternatively, disparity in transformation propensity could be explained by different cells, requiring distinct gene products. In this model, specific oncogenes may only affect selected cell types, as seen in the early development of retinoblastomas when RB is lost[10], or in breast and ovarian cancers when BRCA1 is mutated[11]. But, even when present in the relevant target cell types, oncogenes may not lead to immediate transformation. For example, the chronic myeloid leukemia driver BCR-ABL can persist in hematopoietic stem cells (HSCs) without causing aggressive malignancy[12]. Furthermore, it is conceivable that oncogenic mutations only give rise to malignancy when acquired by rare stem cells. However, when malignancy is manifested by progeny of the mutated stem cells, it is difficult to ascertain whether transformation is initiated in the stem cells themselves or specific types of their differentiated descendants. Indeed, stem cells could even resist transformation as compared to their more differentiated descendents[13]. Overall, the acquisition of malignancy appears to follow yet unappreciated rules.

In this report, we set out to determine the cellular traits that contribute to the acquisition of de novo malignancy. Specifically, we focused on granulocyte–macrophage progenitors (GMPs), which are permissive for MLL fusion oncogene-mediated transformation[7,8]. GMPs expressing an MLL fusion oncogene could produce two types of progeny: differentiated ones despite the oncogene expression, or malignant ones that could eventually develop into lethal acute myeloid leukemia (AML) in vivo. This binary system provides a unique opportunity to dissect the molecular and cellular differences that help to drive malignancy.

## Results

**Tracking single GMPs from normal to malignant.** We used an AML model, for which a single oncogene MLL-AF9 is sufficient to initiate lethal disease[7,8], to unveil potential hidden principles governing the emergence of malignancy. To achieve controlled oncogene expression, we generated an inducible MLL-AF9 allele (iMLL-AF9, iMF9): the cDNA encoding human MLL-AF9 oncogene followed by an IRES-ΔNGFR cassette[14] was targeted into the Hprt locus under the control of a tetracycline response element[15]. This allele was crossed with a constitutively expressed reverse tetracycline transactivator (rtTA) allele[16] (Fig. 1a) to enable doxycycline (Dox)-inducible MLL-AF9 expression, which could be monitored by the coexpressed ΔNGFR on cell surface. As the targeted X chromosome locus differs in copy number between male and female animals, we first compared transgene inducibility in both sexes. As expected from X chromosome inactivation in female cells, GMPs from homozygous iMF9/iMF9 females showed similar Dox-dependent transgene induction as those isolated from iMF9/Y males (Supplementary Fig. 1a–c), and correspondingly displayed comparable expression of key MLL-AF9 target genes, such as Hoxa9 (Supplementary Fig. 1d). Thus, all experiments were performed using homozygous females or males for the iMLL-AF9 allele. This iMLL-AF9 allele eliminates

variability in oncogene copy number or integration sites introduced via viral transduction[7,8,14]. Further, precisely timed Dox addition enables assessment of cellular states before and after oncogene induction.

We detected transformation based on serial colony formation in methylcellulose, a well-accepted surrogate assay for hematopoietic malignancies[9], and validated the pathology in vivo. As expected, MLL-AF9 led to serial colony formation in vitro from GMPs (Fig. 1b). Serial colony formation was dependent on sustained MLL-AF9 expression, as Dox removal greatly decreased proliferation in liquid cultures accompanied with reduced colony formation in methylcellulose (Supplementary Fig. 1e–g), consistent with a similarly targeted MLL–ENL (eleven nineteen leukemia) model[17]. In vivo, 12,000 iMLL-AF9 GMPs induced lethal AML in all animals tested (Fig. 1c, Supplementary Fig. 2a–d), confirming that iMLL-AF9 induction could transform GMPs de novo.

However, bulk cultures obscure cellular heterogeneity with regard to transformation permissiveness. We therefore plated iMLL-AF9 GMPs in micro-wells to visualize individual cells, as well as their progeny (Fig. 1d, Supplementary Fig. 3a–c). Cells in micro-wells proliferated at a comparable rate to those in bulk cultures (Fig. 1e) and formed compact colonies when methylcellulose was added (Fig. 1d). To confirm transformation, primary colonies arising from single iMLL-AF9 GMPs in micro-wells were plucked and replated for secondary colony formation (Supplementary Fig. 3a), the presence of which was scored as an event of malignant transformation.

To improve the throughput of this serial-replating strategy that tracks malignancy initiation from single cells, we attempted to eliminate the pluck-and-replate steps by reducing the formation of nonmalignant primary colonies to negligible levels. We accomplished this by culturing GMPs for 2 days before adding methylcellulose (Fig. 1f, Supplementary Fig. 3d). The decrease in colony formation by these briefly cultured GMPs is consistent with their partial differentiation, and not due to insufficient growth factors/cytokines during the 2-day culture, as addition of interleukin 6 (IL6), macrophage colony-stimulating factor (M-CSF), and granulocyte colony-stimulating factor (G-CSF)[18] did not increase their proliferation rate within this time window (Supplementary Fig. 3e) or alleviate the decrease in colony formation following the 2-day culture (Supplementary Fig. 3f). The very few colonies that did form could not support serial replating (0 out of 30; Fig. 1g). In contrast, many colonies emerged from the 2-day-cultured iMLL-AF9 GMPs when Dox was added (Fig. 1f), the great majority of which supported serial replating (86%, 74 out of 86; Fig. 1g), and upregulated Hoxa9 and Meis1 (Fig. 1h), two well-established MLL-AF9 target genes[19,20]. Their ability to support serial replating and to upregulate MLL-AF9 target gene expression demonstrate that the majority of the methylcellulose colonies developed from single iMLL-AF9 GMPs following a 2-day culture were transformed. Overall, these results indicate that the changes in cellular states during the brief culture renders GMPs to forfeit their colony-forming potential, which is preserved by the induced MLL-AF9 during this time. These results suggest that the molecular changes occurred during the brief culture could help to define the cellular states from which MLL-AF9 initiates transformation de novo.

This modified colony-forming assay enabled us to clonally track hundreds of individual GMPs, from their initial cellular states to when they displayed de novo malignant phenotypes (Supplementary Fig. 3a, Fig. 1g, h). We determined that among the GMPs isolated by surface marker expression[21], only ~25% acquired malignancy (Fig. 1f), even though the same oncogenic cassette was similarly driven by Dox (Supplementary Fig. 1b). Therefore, this experimental system provided the opportunity to

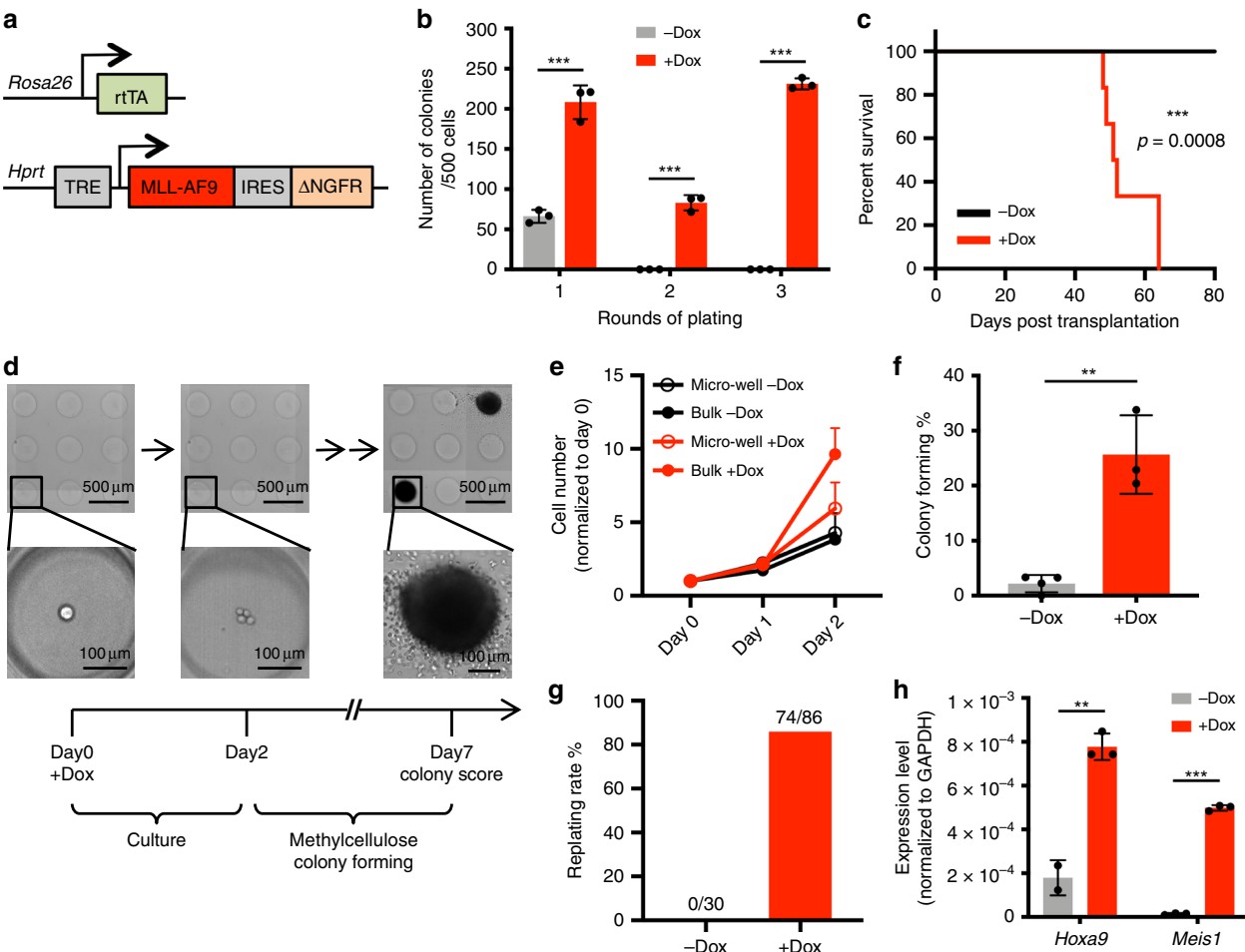

**Fig. 1 Tracking MLL-AF9-mediated transformation from single hematopoietic cells. a** Schema of the inducible MLL-AF9-IRES-ΔNGFR allele targeted into the endogenous *Hprt* locus. **b** Dox-dependent serial colony formation by iMLL-AF9 GMPs; n = 3 for both conditions, p < 0.001 for all comparisons. **c** Kaplan–Meier curve of cohorts of mice transplanted with 12,000 iMLL-AF9 GMPs, fed with control or Dox water (n = 6 for each group). AML pathology was confirmed and shown in Supplementary Fig. 2a–d. p = 0.0008 as calculated by log-rank (Mantel–Cox) test. **d** Representative images tracking single GMPs and their progeny in micro-wells. Cells in micro-wells were kept in liquid culture for 2 days, after which the medium was replaced by methylcellulose (MethoCult GF M3434) to allow colony formation. The presence of colonies was verified after another 5 days (for a total of 7 days in culture). Scale bars represent 500 μm for upper panels, and 100 μm for lower panels. **e** Comparison of iMLL-AF9 GMP proliferation rates in micro-wells and in bulk culture. The proliferation rates in micro-wells were calculated by adding up all cell numbers from individual wells, n = 3 for micro-well culture; n = 2 for bulk culture. **f** Colony forming efficiencies from single iMLL-AF9 GMPs following the schema in **d**. The presence of single cells was confirmed at day 0; n = 4 for −Dox, n = 3 for + Dox, p = 0.0012. **g** Single colonies, as shown in **d**, were plucked and replated in methylcellulose for secondary colony formation. The percentages of colonies that gave rise to replatable colonies are plotted. None of the 30 colonies formed in the absence of Dox supported replating, while 74 of the 86 colonies formed in the presence of Dox supported replating. **h** RT-QPCR analyses of *Hoxa9* and *Meis1* in colonies formed by single iMLL-AF9 GMPs +/−Dox; n = 3, p = 0.002 for *Hoxa9*, p < 0,001 for *Meis1*. Results are presented as mean±s.d. p values (except for Kaplan–Meier curve) were calculated by two-sided unpaired t-test.

compare the cell states prior to oncogene expression and relate that to their future fate outcome.

**Transformation initiates from naturally fast-cycling cells.** We then asked whether the subset of GMPs acquiring malignancy were random, or possessed specific cellular trait(s). We focused on their cell cycle rate, as cell cycle is a major contributor to cellular heterogeneity[22]. Since ultrafast cell cycle is associated with cell fate plasticity[23], we hypothesized that the transformation-permissive GMPs would display a distinct cell cycle rate. Alternatively, if malignancy arises randomly, the cell cycle rate of the transformation-permissive GMPs would resemble that of the bulk GMPs.

The cell cycle rate of single GMPs was determined by directly scoring the number of its progeny at 24 and 48 h (Fig. 1d). Most

GMPs divided 0 to 3 times within a 24-h time widow (Fig. 2a), a cell cycle rate consistent with our previous measurements[23]. The cycling rate during the second 24-h time window did not correlate with that of the first 24 h (Fig. 2a), indicating that the cell cycle rate of individual GMPs are dynamic and hetero-geneous. When MLL-AF9 was induced, however, the cell cycle rate mostly increased during the second 24 h of culture (Fig. 2b). It is important to note that cell cycle rates remained identical between + / −Dox conditions within the first 24 h (Fig. 2c, Supplementary Table 1), diverging only during the second 24 h (Fig. 2d, Supplementary Table 1). Thus, the cell cycle rate during the first 24 h reflects the intrinsic normal GMP behavior, while the second 24 h includes the oncogene effects. The lack of difference in cycling rates during the first 24 h of Dox treatment offers a brief time window to capture the intrinsic GMP cell cycle rate, before it had been significantly altered by the oncogene.

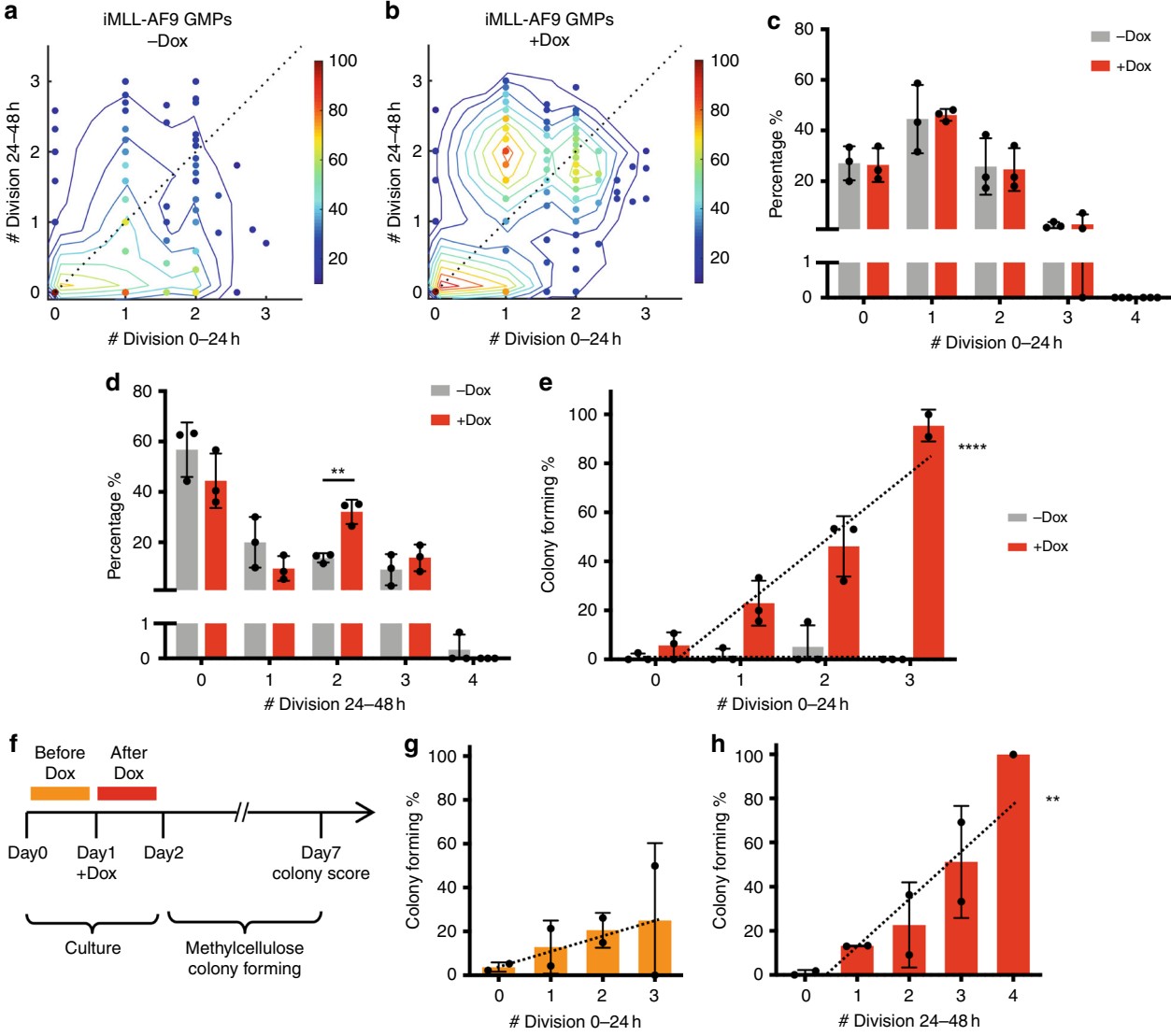

**Fig. 2 MLL-AF9-mediated transformation efficiency is associated with the intrinsic cycling rate of GMPs. a**, **b** Cell cycle rates of individual iMLL-AF9 GMP lineages. The presence of single GMP was confirmed at 0 h, and plotted as individual dots. **a** −Dox: $n = 268$; **b** +Dox: $n = 378$ (Dox was added at 0 h). The number of divisions during 0–24 h (x-axis) = Log2(cell number at 24 h/cell number at 0 h); the number of divisions during 24–48 h (y-axis) = Log2 (cell number at 48 h/cell number at 24 h). Pseudo color denotes GMP lineage density, with the highest observed density in each sample normalized to 100. Dotted diagonal line denotes $y = x$. In the presence of Dox, significant number of GMP lineages underwent cell cycle acceleration, shown as the dense population above the diagonal line. **c** Cell cycle rate distribution during 0–24 h of iMLL-AF9 GMP culture +/−Dox. Cell cycle rates were determined as described in **a**, **b**, and rounded up to the nearest integer. Detailed measurements are shown in Supplementary Table 1; $n = 3$ for +/−Dox measurements. No significant difference was observed between +/−Dox conditions. **d** Similar analysis as in **c** performed for 24–48 h of culture; $n = 3$ for +/−Dox measurements. $p = 0.0036$ for two divisions, as calculated by unpaired t-test. **e** Single-cell colony-forming efficiency by iMLL-AF9 GMPs of different cycling rates as defined by number of cell divisions during 0–24 h in vitro culture. Dotted lines represent linear regressions; $n = 3$ for +/−Dox. For + Dox condition, $y = 28.06 \times -0.4914$, $p < 0.0001$; for −Dox condition, $y = 0.08382 \times +1.778$, $p = 0.9466$. **f** Schematic timeline for measuring colony-forming efficiencies in relation to cell cycle rates before and after Dox addition. Orange bars denote the time before Dox addition and the red bars denote the time after Dox addition. **g** Single-cell colony-forming efficiency in iMLL-AF9 GMPs of different cycling rates prior to Dox induction, as shown in orange in **f**; $n = 2$. Dotted line represents linear regression: $y = 7.140 \times + 4.852$, $p = 0.1998$. **h** Single-cell colony-forming efficiency in iMLL-AF9 GMPs of different cycling rates following Dox induction, as shown in red in **f**; $n = 2$. Dotted line represents linear regression: $y = 21.47 \times -7.473$, $p = 0.0016$. Results are presented as mean±s.d. $p$ value for linear regression was calculated by F-test to assess whether the slope is significantly nonzero.

With the ability to clonally track individual GMP cell cycle rates, we asked whether the intrinsic cell cycle rate relates to its transformability. Indeed, the intrinsic cell cycle rate strongly correlated with their ability to form transformed colonies (Fig. 2e). Almost all GMPs that divided three times or more within the first 24 h (~3% of total GMPs) were transformed. The cell cycle rate of the second 24 h of Dox treatment also correlated with transformation probability (Supplementary Fig. 3g), although

this cell cycle rate could be consequent to the prolonged oncogene activity (Fig. 2d). These data suggest that malignancy does not arise from random GMPs. Rather, a subset of GMPs with the most rapid cell cycle appear more probable to acquire malignancy.

The proliferativeness during 0–24 h could mark select GMP lineages for transformation, or provide the immediate cell states for transformation. To distinguish these possibilities, we delayed

Dox addition for 24 h (Fig. 2f) and assessed the cell cycle rates before (0–24 h) and within the initial 24 h of Dox treatment (24–48 h). Notably, the cell cycle rates prior to Dox treatment (0–24 h) were minimally relevant for transformation outcome (Fig. 2g). Instead, a much stronger correlation between the cell cycle rate at the time of Dox treatment (24–48 h) and their transformation potential was detected (Fig. 2h). These data indicate that transformation permissiveness depends more on the current proliferative state when MLL-AF9 is expressed. This experiment also revealed the emergence of an even faster dividing subsets, which divided four times during the 24–48 h and formed transformed colonies with essentially 100% efficiency (Fig. 2h). These results suggest the possibility that a subset of GMPs undergoing cell cycle acceleration might be particularly permissive to transformation. Taken together, as cell cycle rate undergoes dynamic changes during culture, as likely occurs also during in vivo differentiation, individual cells may become particularly susceptible to transformation when transiting through a highly proliferative state.

**Immediate preservation of a GMP-like state by MLL-AF9**. To understand how the initial cellular state contributes to permissiveness to transformation, we examined the early transcriptomic response to MLL-AF9 induction (Supplementary Fig. 4a). Twenty-four hour of MLL-AF9 induction led to upregulation of genes involved in RNA metabolism and protein translational processes, and downregulation of genes related to immune functions (Supplementary Fig. 4b). Surprisingly, the HSC self-renewal program known to be reactivated by MLL-AF9[8] was not detected at this time. Of the 41 L-GMP signature genes, only 3 were upregulated in the +Dox GMPs (Supplementary Fig. 4c). Several key MLL-AF9 target genes (e.g., *Hoxa9* and *Meis1*) were indeed higher in the +Dox cells (Fig. 3a). However, their absolute expression levels were more comparable to that of freshly harvested GMPs and much lower than that in hematopoietic stem and multipotent progenitors (Lin−c-Kit + Sca-1 +; LKS; Fig. 3a). Thus, the initial gene expression changes induced by MLL-AF9 during this brief culture lack the extensive reactivation of the stem cell program. Additionally, the expression of Dox-downregulated genes was similarly low in fresh GMPs, such as *Acod1* and *Cxcl2*, indicating that Dox induction maintained their low expression level (Fig. 3b). Overall, these data suggest that MLL-AF9 helps preserving the expression level of several genes to that of the originating GMPs. The preservation of a GMP-like gene expression state by MLL-AF9 was widespread (Fig. 3c). Moreover, on absolute gene expression levels, Dox upregulated differentially expressed genes (DEGs) were expressed at higher levels in fresh GMPs, while Dox downregulated DEGs were expressed lower in fresh GMPs (Fig. 3d). Therefore, in the presence of MLL-AF9, an actively expressed gene is likely to retain its high expression while a lowly expressed gene continues to remain low. Of note, the changes in the expression of cell cycle genes were also mitigated by MLL-AF9 (Fig. 3e). Such an effect could lead to a persisting high cell cycle rate in fast-cycling GMPs, potentially contributing to MLL-AF9-mediated transformation (Fig. 2e).

The changes in gene expression were corroborated by corresponding changes on chromatin accessibility. Increased ATAC-seq signals were detected at Dox upregulated genes, such as *Hoxa9* (Supplementary Fig. 4d). Conversely, decreased ATAC-seq signals were found at downregulated genes, such as *Acod1* (Supplementary Fig. 4e). The differentially accessible chromatin regions returned similar Gene Ontology (GO) terms (Supplementary Fig. 4 f) as those of the DEGs (Supplementary Fig. 4b). Importantly, the genomic regions corresponding to the Dox-increased ATAC-seq peaks were already accessible in normal fresh GMPs, while regions corresponding to Dox-decreased ATAC-seq peaks had low accessibility in normal GMPs (Fig. 3f, Supplementary Fig. 4g). Similar to its effect on gene expression, MLL-AF9 appears to prevent culture-induced chromatin region opening (Fig. 3g, left panel), and enhance the accessibility of those already open regions (Fig. 3g, right panel), particularly around its direct target sites[24] (Fig. 3h, i). Overall, upon MLL-AF9 induction, the cell state as represented by gene expression and chromatin openness, retains resemblance to that of the naturally existing GMPs. Viewed in this light, MLL-AF9 could transform cells by preserving and perpetuating an already rapidly proliferating cell state occupied by a subset of normal GMPs.

**A GMP subset naturally express transformation-related genes**. To parse out cellular heterogeneity, we performed single-cell RNA-sequencing (scRNA-seq) of fresh GMPs and GMPs cultured for 1 day in the presence or absence of Dox (+/−Dox). Cultured GMPs expressed variable levels of key MLL-AF9 target genes (*Hoxa9, Meis1,* and *Mef2c* ; Fig. 4a), cell cycle genes (*Cdk4* and *Cdk6* ; Supplementary Fig. 5a), and differentiation related genes (*Spi1* and *Cebpb* ; Supplementary Fig. 5b). The range of gene expression variability is similar between +/−Dox GMPs, although the average expression level was higher in +Dox cells. Therefore, the changes observed by bulk RNA-seq should reflect a cell composition change. Importantly, variability in the expression of these genes did not coincide with expression level of the transgenic cassette (Supplementary Fig. 5c), suggesting that the differential efficiency in transformation should not have been primarily imparted by oncogene dosages.

To determine the cell states in freshly isolated GMPs and GMPs cultured +/−Dox for 24 h, we used PHATE[25], an algorithm optimized for inferring relationships between cellular states from scRNA-seq data. Examination of a PHATE embedding reveals that freshly isolated GMPs deviated greatly from cultured GMPs (+/−Dox; Supplementary Fig. 5d), indicating that the culture-induced gene expression changes dominate at this time. Surprisingly, GMPs cultured +/−Dox displayed largely overlapping population structure (Fig. 4b) with two main diverging trajectories, likely corresponding to granulocyte- or monocyte-committed progenitors, as determined by the level of *Ly6c2* and *Csf1r* (Supplementary Fig. 5e).

We used the MELD package to identify effect of transgene induction on individual cell states[26]. The algorithm uses Enhanced Experimental Signal (EES) to quantify the likelihood ratio of observing a given cell state in either experimental condition. A high EES value corresponds to cell states that are more likely to be observed in the experimental condition relative to the control and vice versa. The distribution of EES across cells from the −Dox and +Dox conditions revealed that the *Ly6c2*-low / *Csf1r*-low subset was enriched after Dox induction (Fig. 4c), corroborating impaired GMP differentiation in the presence of MLL-AF9 (Fig. 3b, Supplementary Fig. 4b).

To identify cell states that were the most enriched or depleted after Dox induction, we applied Vertex Frequency Clustering[26] to groups cells. We identified five clusters of cells in the cultured GMPs based on the expression levels of *Ly6c2* and *Csf1r* (Fig. 4d, e). Clusters 1 and 5 expressed lowest levels of both *Ly6c2* and *Csf1r*, cluster 3 is distinguished by the presence of *Ly6c2* and absence of *Csf1r*, and clusters 2 and 4 express both *Ly6c2* and *Csf1r* (Supplementary Fig. 5e). Notably, cells in clusters 1 and 5 expressed the highest levels of cell cycle genes *Cdk4* and *Cdk6* (Fig. 4f, Supplementary Fig. 5f), as well as key target genes of MLL-AF9 (*Hoxa9*, *Meis1*, and *Mef2c*; Fig. 4g). In contrast, expression of the cell cycle genes and MLL-AF9 target genes were lowest in clusters 2

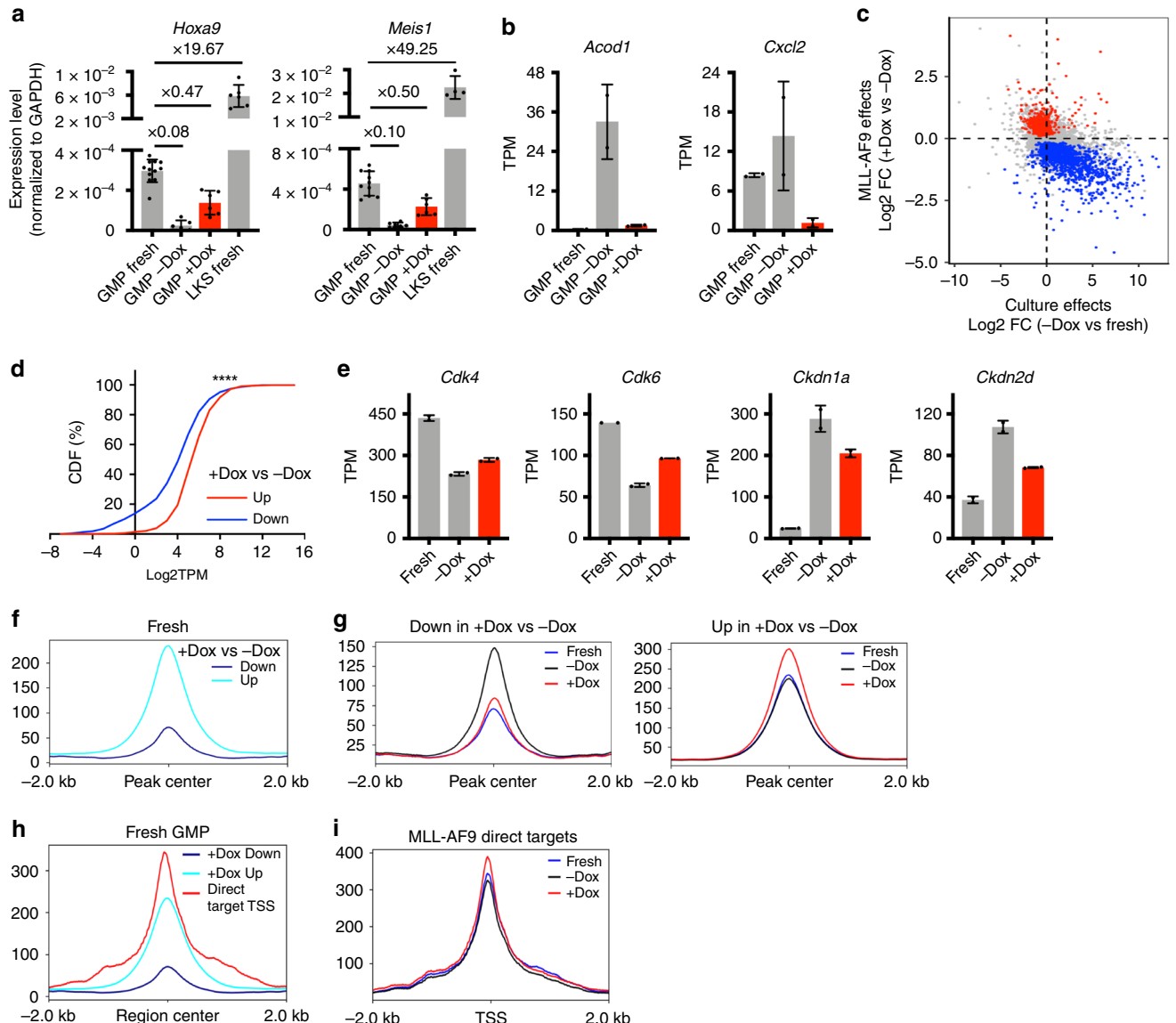

**Fig. 3 MLL-AF9 expression sustains the already existing gene expression program in GMPs. a** The expression levels of *Hoxa9* and *Meis1* in LKS, fresh GMPs, and GMPs cultured +/−Dox. The fold of change in each gene expression relative to the levels in fresh GMPs are shown. $p < 0.01$ for all pair-wise comparisons as calculated by two-sided unpaired *t*-test. **b** Expression levels of *Acod1* and *Cxcl2* in fresh GMPs and GMPs cultured +/-Dox. **c** Scatter plot showing culture-induced gene expression changes (−Dox vs fresh GMPs) vs MLL-AF9-induced changes (+Dox vs −Dox GMPs). Blue dots represent the oncogene downregulated DEGs; red dots represent the oncogene upregulated DEGs. **d** Cumulative distribution function plot showing the expression levels of DEGs in fresh GMPs. DEGs are derived by comparing +/−Dox GMPs. Red line: genes upregulated in +Dox GMPs ($n = 1258$); blue line: genes downregulated in +Dox GMPs ($n = 1715$). TPM: transcript per million. $p < 0.0001$ from Kolmogorov–Smirnov test. **e** Representative expression levels of up/downregulated cell cycle genes. **f** Meta plot summarizing the intensities of up/down ATAC-seq peaks in fresh GMPs. The up and down ATAC-seq peaks are defined by comparing +Dox GMPs vs −Dox GMPs. **g** Dox inductions (red line) prevented the opening of chromatin regions during GMP culture (left, blue vs black lines), and promoted the further opening of chromatin regions that are already accessible in fresh GMPs (right, blue line). The ATAC-seq peaks (down or up) are defined by comparing +Dox GMPs vs −Dox GMPs. **h** The direct targets of MLL-AF9 (red line, defined by MLL-AF9 ChIP-Seq[24]) were highly accessible in fresh GMPs, beyond those more accessible regions caused by Dox induction (cyan line). **i** 24 h of Dox induction (red line) promoted further opening of the genomic regions directly bound by MLL-AF9 (defined by MLL-AF9 CHIP-Seq[24]), which display significant accessibility in fresh GMPs (blue line). Results are presented as mean±s.d.

and 4 (Fig. 4f, g). Importantly, the high expression of *Hoxa9/Meis1/Mef2c* in +Dox GMPs (clusters 1 and 5) was present in similar levels in −Dox GMPs, though population frequencies changed (Fig. 4g). These results demonstrate that a subset of GMPs expressing high levels of *Cdk4/6* and *Hoxa9/Meis1/Mef2c* exist independent of Dox treatment. When such a gene expression program is preserved by MLL-AF9, the emergence of malignant cells becomes highly probable.

**MLL-AF9 preserves gene expression in multiple cell types**. To further assess whether MLL-AF9 preserves gene expression programs in general, we induced its expression in additional cell types. Specifically, we treated LKS cells, GMPs, and differentiated myeloid cells (Mac1+) with Dox for 2 days (Fig. 5a, Supplementary Fig. 6). While the transgenic cassette was similarly induced in all cells (Supplementary Fig. 7a), Dox led to varied proliferative responses (Fig. 5b, Supplementary Fig. 7b, c). GMPs,

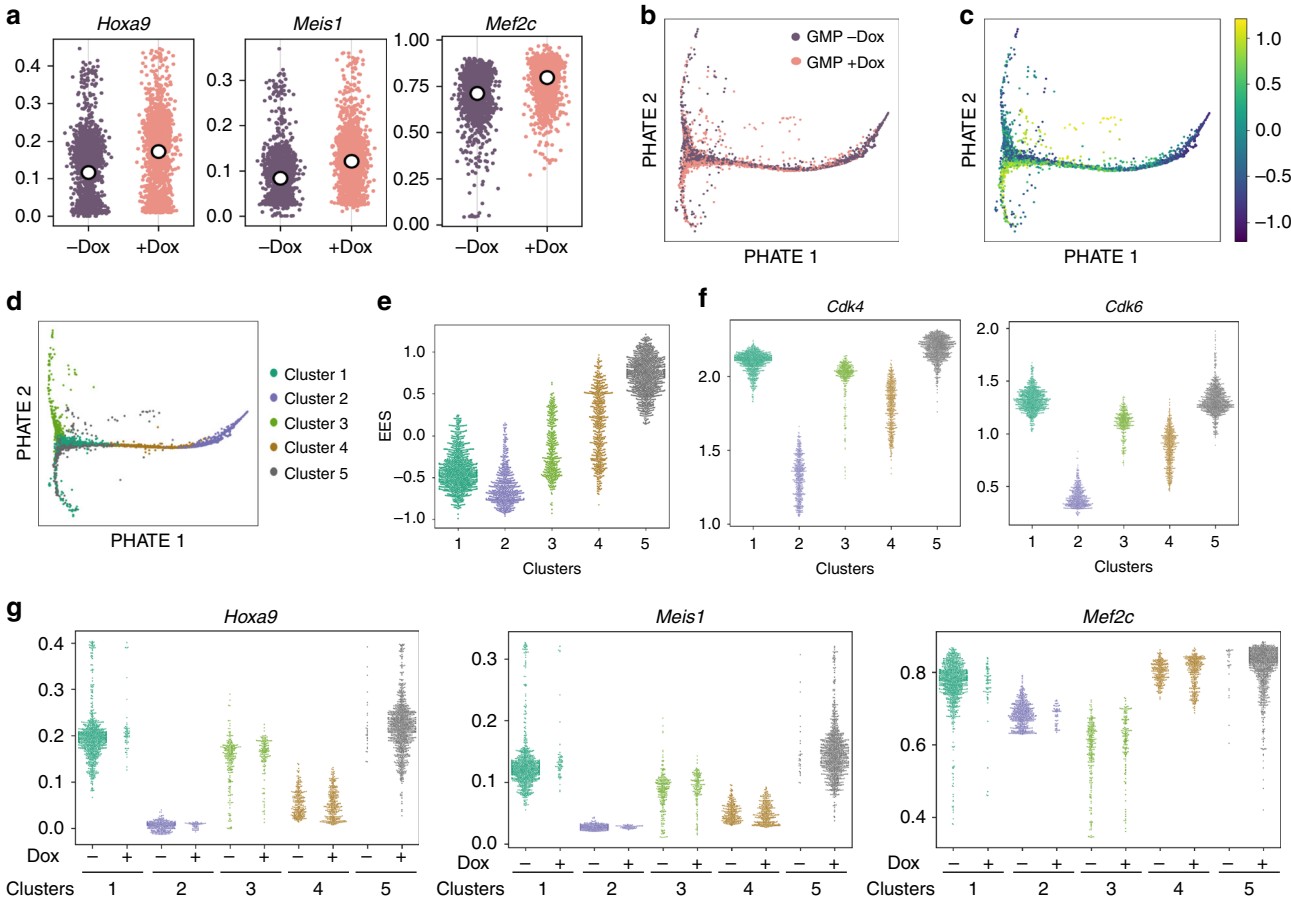

**Fig. 4 MLL-AF9 preserves the heterogeneous gene expression states as analyzed by scRNA-seq. a** Swarm plots showing the expression levels of *Hoxa9*, *Meis1*, and *Mef2c* in cultured GMPs in the presence or absence of Dox. White dots indicate mean of each sample. **b** PHATE plot of cultured GMPs in the presence (pink dots) or absence of Dox (purple dots). **c** Pseudocolor denoting enhanced experimental signal (EES) is overplayed on the PHATE plot. EES is derived by comparing the GMPs cultured in the presence or absence of Dox, where a larger EES score indicates more enrichment in the +Dox GMPs. **d** Based on their gene expression similarity, five clusters of cells were determined. Their positions on the PHATE plot are shown. **e** EES of individual cells within each cluster. **f** Expression levels of *Cdk4* and *Cdk6* in each GMP cluster. **g** Expression levels of *Hoxa9*, *Meis1*, and *Mef2c* in each GMP cluster. The values for cells cultured in the absence or presence of Dox were plotted separately.

which are naturally proliferative[27], tend to increase proliferation and cell cycle gene expression in the presence of Dox, while neither LKS nor differentiated myeloid cells did so, coinciding with their slower or lack of proliferation. Furthermore, although Dox failed to enhance proliferation of the largely quiescent fresh LKS cells (Supplementary Fig. 7d), it did so when LKS were first activated into cycling by culture[23] (Supplementary Fig. 7e). The differential responses in fresh and cultured LKS cells further support the importance of initial cellular states in determining MLL-AF9 effects. These data are consistent with a previous report for MLL–ENL-expressing HSCs[13], suggesting functional conservation among MLL fusion oncogenes. Taken together, MLL-AF9 expression does not impart a universal proliferative response (Fig. 5b, Supplementary Fig. 7b–e). Rather, such an effect requires the starting cells to be already proliferative.

In agreement with the disparate proliferative responses upon oncogene induction, there was minimal overlap amongst the Dox-induced DEGs across the three cell types (Fig. 5c). One of the five commonly upregulated genes is *Hprt* (Supplementary Fig. 7f), the host locus for the iMLL-AF9 allele (Fig. 1a), corroborating successful transgene induction in all cell types (Supplementary Fig. 7a). Taken together, identity of the primary MLL-AF9-responsive genes differ according to the specific cell types in which it is expressed. An early universal gene signature induced by MLL-AF9 was not present. However, MLL-AF9 did

invariably counter the culture-elicited gene expression changes (Fig. 5d) in all cases. Specifically, as observed for GMPs, MLL-AF9 upregulated expression of genes enriched in their respective freshly isolated counterparts, while downregulating those negatively enriched in the fresh cells (Fig. 5e). The enrichment was specific to the respective cell type itself and no enrichment across cell types was detected (Supplementary Fig. 7g). Thus, instead of inducing a common gene signature, MLL-AF9 sustains the existing gene expression programs in multiple cell types.

**Cell cycle modulation alters GMPs' transformation efficiency.** Given that MLL-AF9-mediated transformation is strongly associated with rapid GMP proliferation, and MLL-AF9 preserves cellular states, modifying cell cycle rate of the initiating cells could potentially change the potency of MLL-AF9 to induce transformation. To test this possibility, we treated GMPs with palbociclib (PD0332991), a CDK4/6 inhibitor[28]. At 500 nM, palbociclib mildly and reversibly decreased GMP proliferation (Supplementary Fig. 8a–c), and preferentially abrogated the most proliferative GMP subsets, with the cells that could divide three times or more within 24 h becoming barely detectible (Supplementary Fig. 8d, Supplementary Table 2). Tracking the cells in micro-wells revealed that palbociclib treatment decreased GMPs' ability to form transformed colonies (Fig. 6a, b). Palbociclib treatment of

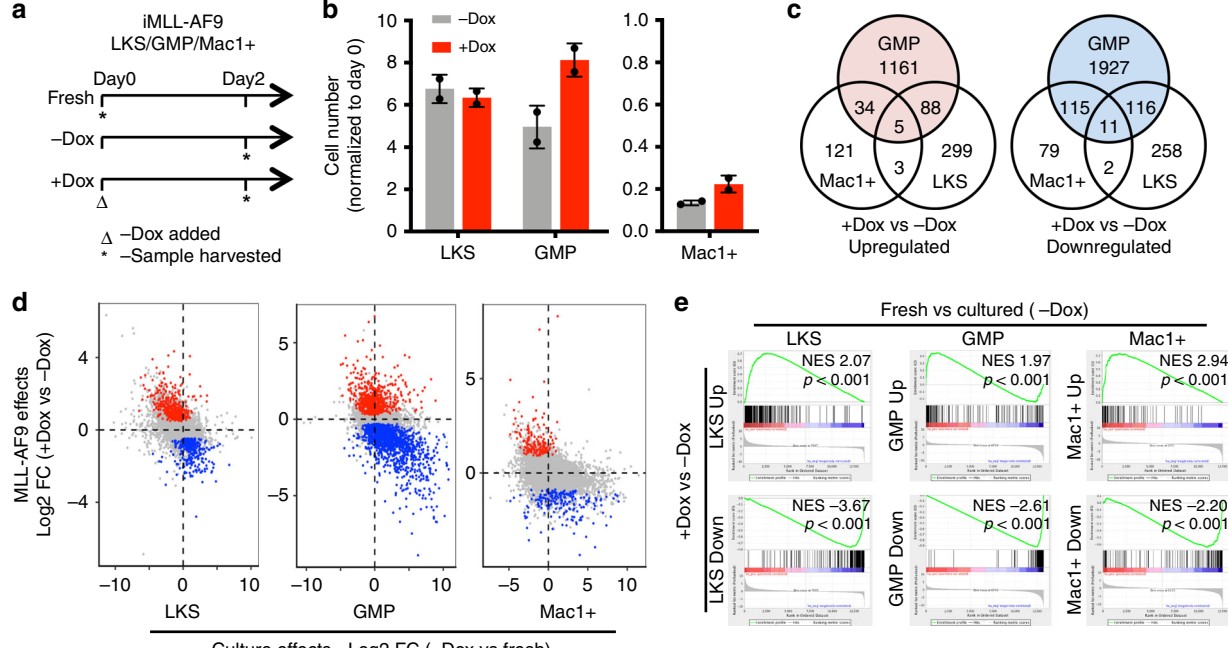

**Fig. 5 The primary gene expression changes in response to MLL-AF9 induction is cell context dependent. a** Workflow showing iMLL-AF9 LKS, GMP, or Mac1+ cells after their isolation on day 0 (fresh), or treated +/−Dox for 2 days. **b** Proliferation rates of respective cell types during 2 days of culture +/−Dox; $n = 2$, error bars represent s.d. **c** Venn diagram showing little overlap among up/downregulated DEGs from LKS, GMP, and Mac1+cells. Cell type-specific DEGs ($p < 0.05$) were defined between the respective cells treated +/−Dox. **d** Scatter plot showing culture-induced gene expression changes (−Dox vs fresh) vs MLL-AF9-induced changes (+Dox vs −Dox) in three cell types. Blue dots represent the oncogene downregulated DEGs; red dots represent the oncogene upregulated DEGs. **e** The Dox-induced DEGs, similar to those defined in **c** but at $p < 0.01$, were queried against the gene expression changes occurred during culture in the respective cell types by GSEA. For GMPs, only the top 200 DEGs were used.

iMLL-AF9 GMPs also significantly decreased transformed colony formation in bulk methylcellulose cultures by ~50% (Fig. 6c, Supplementary Fig. 8e, f), where colony size is not restricted by the micro-well and could serve as an indicator of overall proliferativeness. Importantly, the size of the persisting transformed colonies did not decrease (Fig. 6d, Supplementary Fig. 8g), suggesting palbociclib reduced the units of transformation, but not the general proliferative capacity of the still transformation-competent cells.

To examine whether the palbociclib-modified GMP state decreases transformation in vivo, we transplanted 20,000 iMLL-AF9 GMPs, a dosage where all recipient mice developed AML when fed with Dox (Fig. 6e). Cohorts of Dox-fed mice were briefly treated with either vehicle control or palbociclib at a previously validated dosage[29], immediately following iMLL-AF9 GMP injection (Fig. 6e, Supplementary Fig. 8h). Similar to the results in vitro, this transient palbociclib treatment reduced transformation in vivo, leading to decreased leukemia burden and prolonged animal survival (Fig. 6f–h). A subset of the palbociclib-treated mice never developed AML and remained healthy for the entire time studied (Fig. 6h). Together, these results demonstrate that transient cell cycle inhibition reduces the likelihood of malignant transformation in vivo.

On gene expression level, palbociclib treatment decreased expression of many cell cycle-related genes (Supplementary Fig. 9a–c), but not apoptosis-associated genes (Supplementary Fig. 9d). In the presence of palbociclib, MLL-AF9 led to fewer Dox-induced DEGs (Fig. 6i, Supplementary Fig. 9e–g). Even for those common DEGs, the gene expression fold changes were subdued (Fig. 6j, Supplementary Fig. 9h). Taken together, these data support that even mild cell cycle inhibition of the initiating GMPs could compromise MLL-AF9's oncogenic potential,

consistent with the preservation of cellular states devoid of the most aggressive cell cycle behavior.

Conversely, we tested whether cell cycle acceleration potentiates MLL-AF9-mediated GMP transformation, using a model of emergency myelopoiesis in which GMP cell cycle is further activated by fluorouracil (5FU)[30]. EdU pulse labeling (Supplementary Fig. 10a) and single-cell, cell cycle assay (Fig. 7a) both confirmed enhanced GMP proliferation 8 days post 5FU injection, consistent with the previous report[30]. As expected, this accelerated cell cycle was accompanied by increased transformed colony formation (Fig. 7b). At clonal level, transformed colony-forming efficiency also strongly correlated with cell cycle rates at the time of MLL-AF9 induction (Fig. 7c).

To determine whether leukemogenesis in vivo is potentiated during emergency myelopoiesis, we injected 1000 iMLL-AF9 GMPs, homeostatic or activated with 5FU, into wild-type recipient mice fed with Dox water (Fig. 7d). In agreement with increased colony formation, 5FU-activated GMPs led to higher levels of myeloid engraftment (Fig. 7e, Supplementary Fig. 10b) and markedly reduced disease latency (Fig. 7f). At this limiting cell dosage, 2 out of the 6 recipients that received normal GMPs did not display donor-derived cells and remained leukemia-free (Fig. 7f), In contrast, all recipients (4 out of 4) transplanted with 5FU-activated GMPs quickly succumbed to leukemia (Fig. 7f), with a disease latency similar to animals engrafted with 12,000 homeostatic GMPs (Fig. 1c). These results are in agreement with the increased frequency of GMPs displaying strong cell cycle activation gene signature at day 8 following 5FU injection[30]. Overall, the increased leukemogenesis by the regenerating GMPs further support that transformation by MLL-AF9 is potentiated when more GMPs adopt the highly accelerated cell cycle (Fig. 7a). These results echo the findings using cytokine-induced

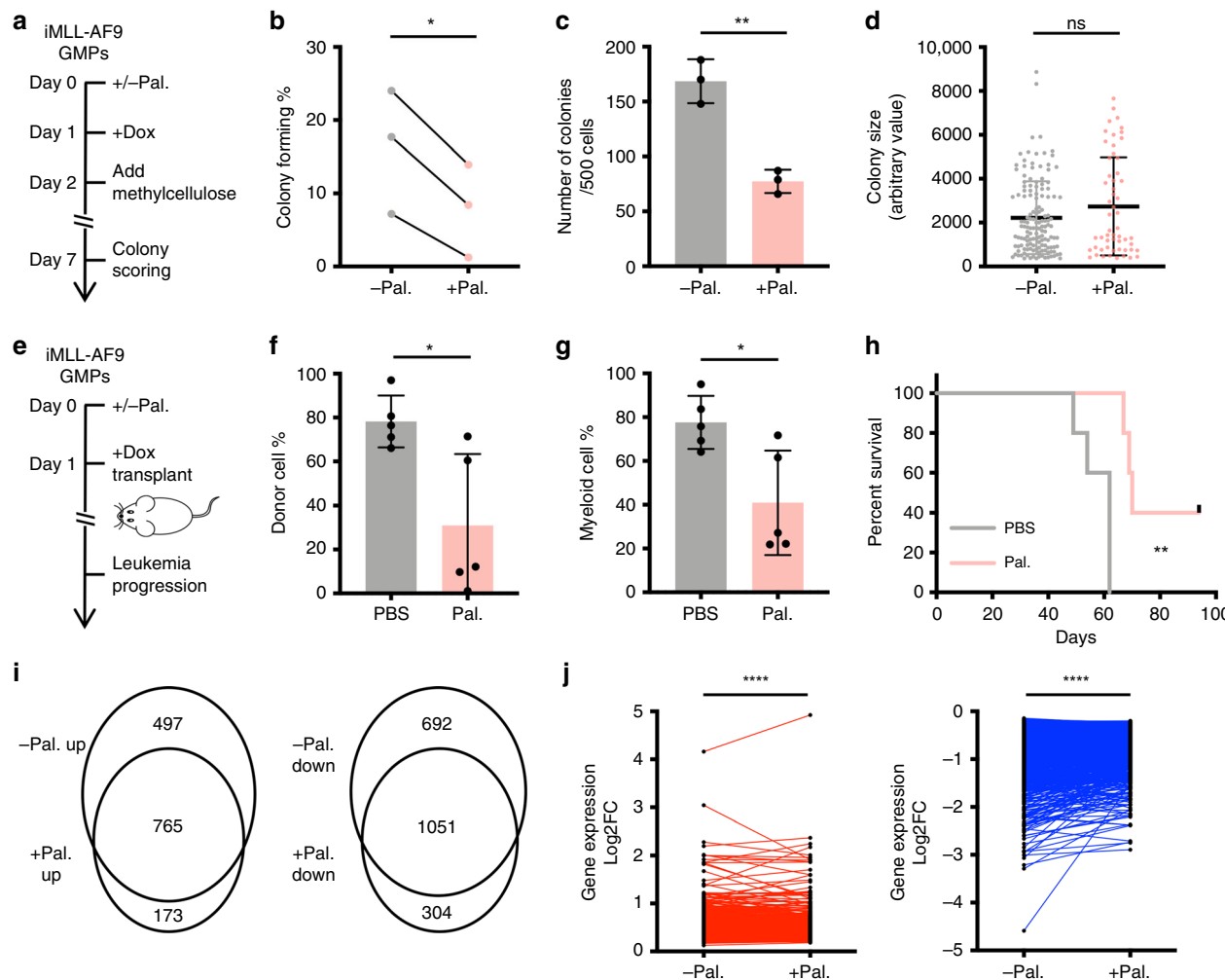

**Fig. 6 Modification of the initial cell state by cell cycle inhibition mitigates MLL-AF9-mediated transformation. a** Schematics illustrating the transformed colony formation by iMLL-AF9 GMPs +/−Pal. **b** Single-cell colony-forming assay showing colony forming efficiencies from iMLL-AF9 GMPs transiently treated with palbociclib. Each pair of dots connected by the line represent colony forming efficiencies of GMPs from a different iMLL-AF9 mouse; ($n = 3$), $p = 0.0221$ by paired $t$-test. **c, d** Quantification of colony number **c** and size **d** of bulk cultured iMLL-AF9 GMPs in methylcellulose +/−Pal. Dox was added to all conditions. To prevent proliferation rate recovery after palbociclib washout (Supplementary Fig. 8c), we maintained palbociclib at 250 nM after treating the cells at 500 nM for the first 2 days (Supplementary Fig. 8e, f). By unpaired $t$-test, $p = 0.0022$ for **c** ($n = 3$); $p = 0.0761$ for **d** ($n = 143$ for −Pal, $n = 56$ for +Pal). **e** Schematics illustrating in vivo leukemogenesis by iMLL-AF9 GMPs. iMLL-AF9 GMPs were treated with palbociclib in vitro for 1 day (+/−Pal.) prior to their injection into sublethally irradiated recipient mice, which received three injections of palbociclib or vehicle control (+/−Pal.) on day 0, day 1, and day 2. All mice received 20,000 GMPs and were fed with Dox water. **f** Percentage of donor-derived cells (CD45.1/2) in the peripheral blood of recipient mice (CD45.2), assayed 7 weeks post transplantation; $n = 5$ for each group, $p = 0.0156$ by unpaired $t$-test. **g** Percentage of myeloid cells (Mac1+) in the peripheral blood of recipient mice, assayed 7 weeks post transplantation; $n = 5$ for each group, $p = 0.0155$ by unpaired $t$-test. **h** Kaplan–Meier curve of recipient mice developing AML after iMLL-AF9 GMP transplantations; $n = 5$ for each group, $p = 0.0034$ by Log-rank (Mantel–Cox) test. **i** Venn diagrams showing the Dox-induced DEGs when palbociclib was present (+Pal.) or not (−Pal.). **j** Dox-induced gene expression changes (Log2 fold change) in the absence or presence of palbociclib. The 765 commonly upregulated DEGs in **i** are shown in red, and the 1051 commonly downregulated DEGs in **i** are shown in blue. Each pair of dots denotes an individual gene and is connected by a line in +/−Pal. conditions. $p < 0.0001$ for both up and downregulated DEGs, $p$ value was calculated by paired $t$-test. Results are presented as mean±s.d.

emergence myelopoiesis[31], and support the notion that modifying the proliferative state of normal cells could alter the probability of leukemogenesis by MLL-AF9.

**Molecular features of MLL fusion leukemia-initiating cells.** To refine the understanding of the cell states permissive for MLL-AF9-mediated transformation free of culture-related artifacts, we dissected the heterogeneity in freshly isolated GMPs. Since, cell surface expression of Ly6C and CD115 (encoded by *Ly6c2* and *Csf1r*, respectively) have been reported to mark three developmental stages (Ly6C−CD115$^{lo}$ for multipotent GMPs, Ly6C + CD115$^{lo}$ for granulocyte-committed GPs, and Ly6C + CD115$^{hi}$ for monocyte-committed MPs)[32,33], we isolated these subsets based on Ly6C and CD115 (Supplementary Fig. 11a). When analyzed using our single-cell tracking assay (Fig. 1d), all three subsets displayed similar cell cycle rate distribution, with the Ly6C + CD115$^{hi}$ MPs having slightly more cells dividing twice within the first 24 h of culture (Fig. 8a). The three subsets, however, displayed distinct cell cycle rates during the second 24 h of culture (Fig. 8b): a large portion of Ly6C−CD115$^{lo}$ GMPs underwent cell cycle acceleration, whereas more Ly6C + CD115$^{hi}$ MPs decelerated, with the Ly6C + CD115$^{lo}$ GPs displaying mild

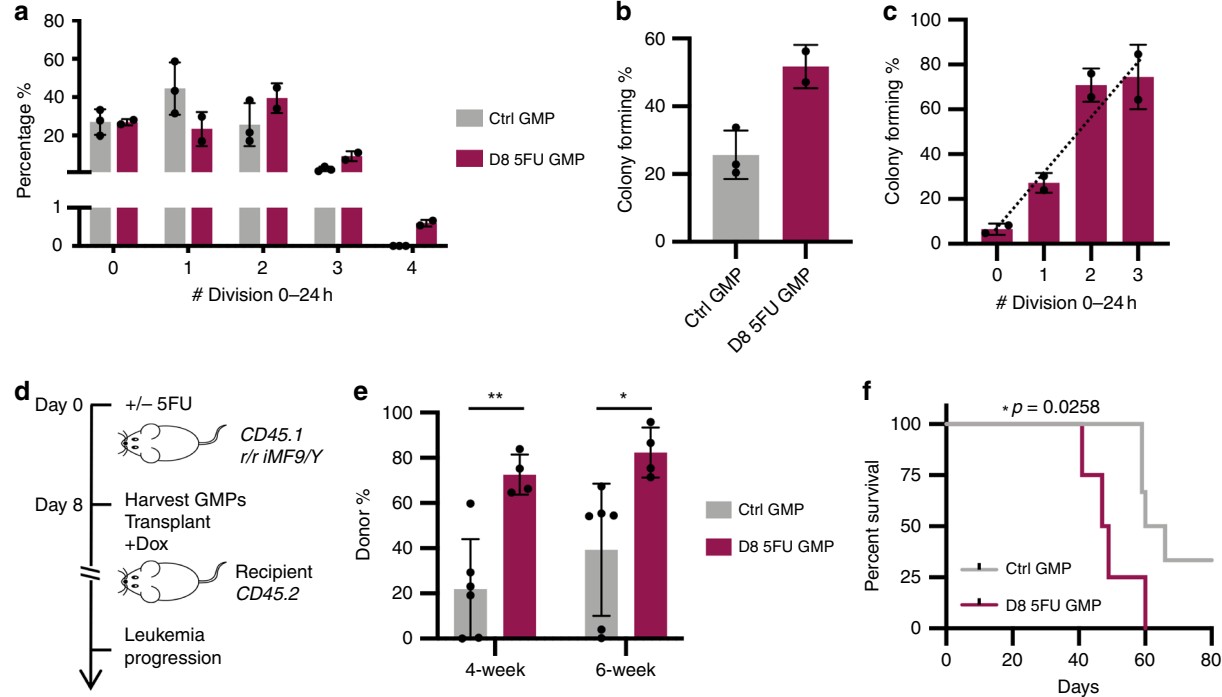

**Fig. 7 Activated GMPs during emergency myelopoiesis support more efficient transformation. a** Cell cycle rate distributions of steady-state control GMPs or 5FU-activated GMPs; $n = 3$ for Ctrl GMP, $n = 2$ for D8 5FU GMP. **b** Single-cell colony-forming efficiencies from steady-state control iMLL-AF9 GMPs or 5FU-activated iMLL-AF9 GMPs; $n = 3$ for Ctrl GMP, $n = 2$ for D8 5FU GMP. **c** Colony-forming efficiencies by 5FU-activated iMLL-AF9 GMPs of different cell cycle rates; $n = 2$, dotted line represents linear regression, $y = 24.73 \times +7.649$, $p = 0.0006$ by F-test. **d** Schematics showing leukemogenesis by steady-state control iMLL-AF9 GMPs or 5FU-activated iMLL-AF9 GMPs. **e** Donor-derived (CD45.1+) nucleated cell percentages in peripheral blood of recipient mice at 4- or 6-weeks after transplantation; $n = 6$ for Ctrl GMP, $n = 4$ for D8 5FU GMP, $p = 0.0027$ for 4 weeks, $p = 0.0243$ for 6 weeks, as calculated by unpaired t-test. **f** Kaplan–Meier curve of the cohorts injected with steady-state control iMLL-AF9 GMPs or 5FU-activated iMLL-AF9 GMPs; $n = 6$ for Ctrl GMP, $n = 4$ for D8 5FU GMP, $p = 0.0258$ as calculated by Log-rank (Mantel–Cox) test. Results are presented as mean±s.d.

cell cycle rate change (Fig. 8c, d). Their cell cycle rate change agreed with their expression of cell cycle genes (Fig. 8e): Ly6C–CD115lo GMPs expressed higher *Cdk4/6* than GPs and MPs. These results demonstrate that subset of GMPs naturally express higher levels of *Cdk4/6* in vivo. Furthermore, since the *Ly6c2loCsf1rlo* cells also express low levels of *Irf8* and *Cebpe* (Fig. 8f, Supplementary Fig. 11b, c), two genes previously reported to mark monocytic and granulocytic commitment[33], they likely represent the metastable GMPs. Taken together, these data strongly suggest that the cell states similar to those represented by clusters 1 and 5 in our scRNA-seq analyses (Fig. 4d–g) occur normally during hematopoietic differentiation. Commitment of the multipotent GMPs to MP or GP states is likely accompanied by further cell cycle acceleration (Fig. 8g), although similar cell cycle rate heterogeneity remains in each subset.

Since cell cycle rate is intimately coupled with developmental stage, we attempted to assess their complex contribution to transformation. If an intrinsically fast cell cycle is key to transformation, a sufficiently fast-dividing MP or GP should still be permissive for transformation despite their more advanced differentiation stage. Indeed, transformed colony formation correlated with cell division rates within each subset (Fig. 8h, Supplementary Fig. 11d). Furthermore, while ~25% of the slow-dividing multipotent GMPs (zero division in 24 h) formed transformed colonies, 70% of the fast-dividing GPs and 50% of the fast-dividing MPs (three divisions in 24 h) transformed, respectively (Fig. 8h). Therefore, a faster-dividing more differentiated cell could be more permissive to transformation than its slower-dividing multipotent GMP predecessor. At least within

this oncogene model, cell cycle rate at the time of oncogene induction appears critical. Differentiation stage also plays a role as GPs appear more permissive to transformation than the similarly dividing MPs, and GMPs gave rise to overall more transformed colonies (Fig. 8h), although the latter could have occurred when individual GMPs transited through a MP- or GP-like state (Fig. 8g). In all three subsets, MLL-AF9 was effectively induced (Supplementary Fig. 11e), arguing against the possibility that differences in permissiveness was driven by deferring oncogene levels.

Lastly, to assess whether a highly proliferative cell state is conserved in humans to drive AMLs, we examined the expression of several cell cycle genes, including *CCNDs* in human GMPs and AMLs, given that D-type cyclins are required for CDK4/6 function[34]. Human GMPs fractionated into three subsets (GMP-A, GMP-B, and GMP-C)[35] displayed different *CCND* expression, with GMP-A being the only subset that expresses detectible level of *CCND1*. GMP-A also expressed higher *HOXA9* and *CDK6* (Fig. 9a), suggesting its resemblance to mouse Ly6C–CD115lo GMPs. In a model where MLL fusion leukemia cells carry on the gene expression program of the cell-of-origin, the resulting leukemia cells should retain similar cell cycle machinery components as the initiating progenitors. Specifically, GMP-A is expected to be more permissive to transformation given their high expression of *CDK6/CCND1* and *HOXA9*. In agreement, while *CDK6* and *HOXA9* are known for their roles in this disease[20,36], we found that *CCND1* expression stratifies MLL patient survival in the TARGET-AML patient cohort[37], with the high expression cases having significantly worse prognosis (Fig. 9b). In contrast, *CCND1* level has no prognostic value in

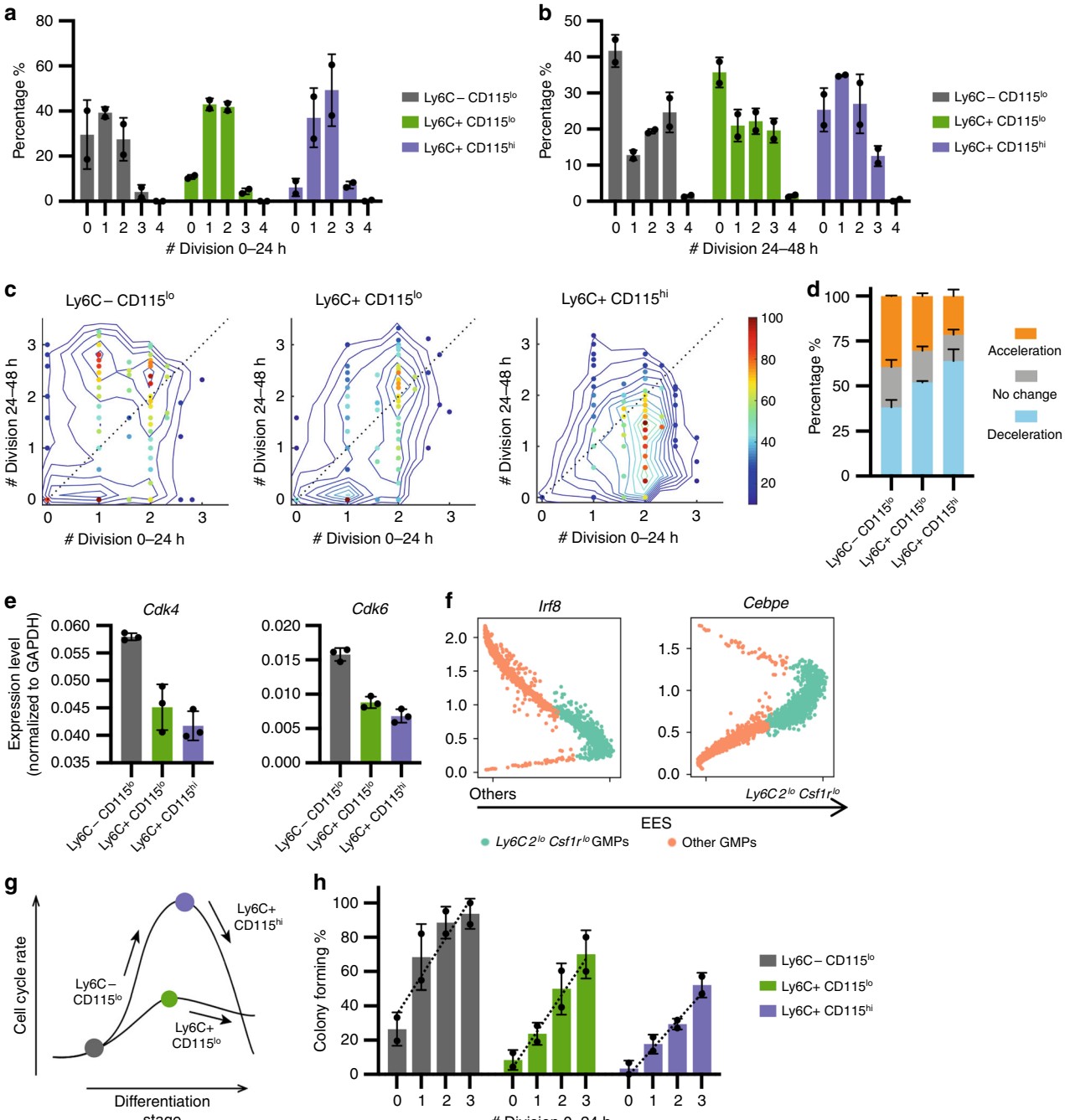

**Fig. 8 Ly6C−CD115$^{lo}$ GMPs are further enriched for MLL-AF9-mediated transformation. a** Cell cycle rate distributions of the GMP subpopulations during 0–24 h of culture, in the absence of Dox ($n = 2$). **b** Cell cycle rate distributions of the GMP subpopulations during 24–48 h of culture, in the absence of Dox ($n = 2$). **c** Cell cycle rates of individual cell lineages during 48 h of culture. Number of divisions during 0–24 h of culture ($x$-axis) were plotted vs those of 24–48 h ($y$-axis). Dotted diagonal line denotes $y = x$. Pseudocolor denotes lineage density, with the highest observed density in each sample normalized to 100. One representative experiment was shown, $n = 161$ for Ly6C−CD115$^{lo}$, $n = 168$ for Ly6C + CD115$^{lo}$, $n = 180$ for Ly6C + CD115$^{hi}$. **d** Quantifications of the proportion of accelerating/constant/decelerating lineages within each GMP subset ($n = 2$). **e** RT-QPCR results showing the expression levels of *Cdk4* and *Cdk6* in FACS-sorted GMP subpopulations based on Ly6C and CD115 ($n = 3$). **f** Expression levels of *Irf8* and *Cebpe* in fresh GMPs from scRNA-seq. Green dots denote the *Ly6c2$^{lo}$Csf1r$^{lo}$* GMPs; orange dots denote the others. $x$-axis represents EES, with the higher EES values indicating more similar gene expression to *Ly6c2$^{lo}$Csf1r$^{lo}$* GMPs. **g** Illustration of cell cycle rate dynamics during GMP differentiation. **h** Transformed colony-forming efficiencies by each subset of GMPs of different cycling rates ($n = 2$). All cells were cultured in the presence of Dox. Dotted lines represent linear regressions, for Ly6C−CD115$^{lo}$, $y = 22.20 × +36.02$, $p = 0.0032$; for Ly6C + CD115$^{lo}$, $y = 21.09 × +6.380$, $p = 0.0004$; for Ly6C + CD115$^{hi}$, $y = 15.80 × +1.933$, $p < 0.0001$. $p$ value was calculated by $F$-test to assess whether the slope is significantly nonzero. Results are presented as mean±s.d.

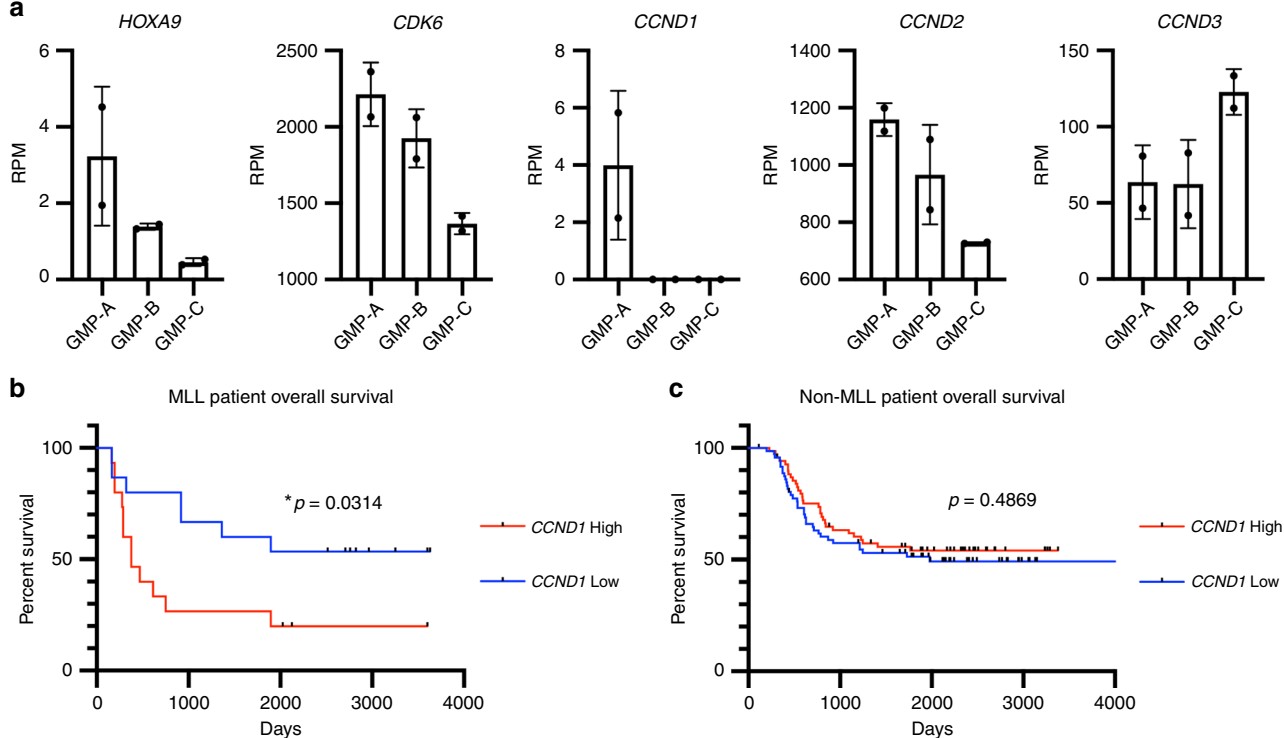

**Fig. 9 CCND1 expression level stratifies MLL translocation leukemia patient survival. a** Expression levels of *HOXA9*, *CDK6*, and *CCND1/2/3* in human GMP subsets. RNA-seq data from GSE96811 were plotted[35]. Error bars represent s.d. **b** Overall survival of MLL leukemia patients. Patient data were downloaded from TARGET-AML datasets, and only patients with available RNA-seq data were included in the analysis. 30 MLL leukemia patients were divided into two groups according to their *CCND1* expression level. 15 patients with *CCND1* level higher than median were grouped as *CCND1* High, 15 patients with *CCND1* level lower than median were grouped as *CCND1* Low, $p = 0.0314$, by Log-rank (Mantel–Cox) test. **c** Overall survivals of nonMLL leukemia patients. Same analysis as described in **b** was performed in nonMLL leukemia patients; $n = 72$ for *CCND1* High group, $n = 71$ for *CCND1* Low group, $p = 0.4869$ by Log-rank (Mantel–Cox) test.

nonMLL AMLs (Fig. 9c). These results support that MLL-rearranged oncogenes likely follow similar mechanisms to drive human AML leukemogenesis, as the one identified here using mouse models.

## Discussion

Our work revealed an important parameter for the normal hematopoietic progenitors that permit MLL-AF9-mediated transformation. Because malignant transformation is a protracted and rare event, it has been difficult to experimentally determine which cells initiate malignancy, among those bearing cancer predisposing mutation(s). We overcame this limitation by establishing an inducible MLL-AF9 allele combined with a system to track transformed fate outcome at clonal level. We identified a cellular state in which the presence of this single oncogene makes transformation nearly a certainty. In this oncogenic model, the proliferative state is not the consequence of malignant transformation, but rather a prerequisite for its initiation. Based on the observation that MLL-AF9 helps to preserve the ongoing gene expression program across multiple cellular contexts, we propose that the rapidly proliferating, immature myeloid progenitor cell state initiates transformation when this cell state is perpetuated by MLL-AF9 expression, fulfilling the functional definition of malignancy. Our data are in agreement with the rapidly cycling myeloid progenitors to be the immediate cell-of-origin[13,31,38,39], and suggest HSCs to be more probable in serving as the reservoir for supplying the rapidly cycling progenitor compartment. The heightened permissiveness to transformation by faster-cycling cells may underlie the myeloproliferative phase and clonal hematopoiesis[40] preceding many AMLs, possibly by increasing the number of cells permissive

of transformation. As the expression of a specific cyclin, *CCND1*, predicts worse survival in MLL fusion AML but not other types of AML, we suggest that a similar mechanism could operate in both mouse and human AMLs initiated by MLL fusion oncogenes (Fig. 9b, c). The prognostic value of *CCND1*, as well as the reduction in leukemogenesis by transient cell cycle inhibition in vivo suggest that targeting cell cycle acceleration might be particularly relevant for preventing this class of AMLs.

Our findings shed light on the vulnerability of this malignancy, and suggest several areas for future investigation. Our model that MLL-AF9 sustains ongoing gene expression programs of fast-cycling cells implies that mitotic bookmarking mechanisms[41,42] might be particularly relevant for MLL fusion oncogenes. MLL can bind to mitotic chromatin in addition to interphase chromatin[43], marking the regions for rapid transcriptional reactivation in the next cell cycle. Future studies should address whether MLL-AF9's effectiveness in fast-cycling cells stems from the oncoprotein's preference for mitotic chromatin. Nonetheless, our model could easily explain why MLL fusion leukemias are sensitive to BET inhibitors[44,45] and CDK4/6 inhibitors[36,46], as BRD4 is a component of the mitotic bookmarking machinery[47] and CDK4/6 inhibition halts G1/S progression[48], respectively. Besides malignancy, the same rapidly proliferating GMPs are also extraordinarily efficient in initiating pluripotency[23], although the mechanisms responsible for these two processes should be distinct. While acquisition of pluripotency represents a departure from the somatic state, MLL fusion oncogene-mediated transformation perpetuates it. This might be the underlying reason why the same Dot1L inhibitor promotes somatic cell reprogramming[49] but inhibits MLL fusion leukemia[50,51].

## Methods

**Mice**. All mouse work has been approved by the Institutional Animal Care and Use Committee (IACUC) of Yale University. All mice used in this study were maintained at Yale Animal Resources Center (YARC). To generate the inducible MLL-AF9 knock-in mouse, the cDNA encoding human MLL-AF9 linked with IRES-NGFR[14] was targeted into the *Hprt* locus in A2Lox.cre mESC cell line[15] by cre-recombination. Correctly, targeted mESCs were injected into E3.5 blastocysts by Yale Genome Editing Center. High-degree male chimeras were crossed with C57BL/6 females to derive the inducible MLL-AF9 knock-in mouse line. The knock-in mice were backcrossed to C57BL/6 background and crossed with a *Rosa26 rtTA* allele[16], and bred to reach homozygosity for both *MLL-AF9* and *rtTA* alleles. Mouse genotyping was determined by polymerase chain reaction (PCR) using primers listed in Supplementary Table 3. This mouse model will be available upon request, before depositing to Jackson Lab.

**FACS sorting and analysis**. Bone marrow cells obtained by crushing mouse tibia, femur, and pelvic bones were incubated with Biotin-conjugated lineage antibodies (CD11b, Ly-6G/6C, Ter119, CD45R/B220, CD3e, CD4, CD8a, 1:140 dilution for each, clone information was listed in Supplementary Table 4), followed by Streptavidin MicroBeads (Miltenyi Biotec, 130-048-101) and LD Columns (Miltenyi Biotec, 130-042-901) to deplete lineage-committed cells. Lin− cells were then stained with CD117-APC (BD Pharmingen, 553356, 1:100 dilution), Sca-1-PE (eBioscience, 12-5981-82, 1:300 dilution), Streptavidin-BV510 (BD Horizon, 563261, 1:300 dilution), CD16/32-PE-Cy7 (eBioscience, 25-0161-82, 1:150 dilution), and CD34-Alexa Fluor 700 (BD Pharmingen, 560518, 1:20 dilution) or CD34-FITC (BD Pharmingen, 560238, 1:50 dilution). GMPs were isolated using fluorescence-activated cell sorting (FACS) on BD Aria as defined previously (Lin−c-Kit + Sca-1−CD34 + CD16/32 +)[21,23]. To fractionate GMPs based on Ly6C and CD115 expression levels, Ly6C-FITC (BD Pharmingen, 561085, 1:150 dilution), and CD115-BUV395 (BD OptiBuild, 743642, 1:200 dilution) were included. Hematopoietic stem and progenitor cells upstream of GMP were sorted as LKS from lineage-depleted bone marrow cells. Differentiated myeloid cells (Mac1+) were sorted as Mac1 + Gr1int (CD11b-PE, BD Pharmingen, 553311, 1:800 dilution; Ly6G/Ly6C-FITC, BD Pharmingen, 553126, 1:200 dilution) from Lin+ cells recovered from LD Columns.

FACS analyses were done using BD LSRII and analyzed with FlowJo. Antibodies used for analyses were the same as those for cell sorting. For CD45.1/CD45.2 staining, CD45.1-BV711 (BD Horizon, 563982, 1:200 dilution) and CD45.2-Pacific Blue (BioLegend, 109820, 1:500 dilution) were used.

For Hochest DNA content staining, cells were fixed in 70% ethanol for 30 min on ice, followed by staining with 5 μg/mL Hochest at room temperature for 15 min, and analyzed on LSRII directly.

**Cell culture**. Freshly isolated GMPs were cultured in x-vivo15 (Lonza, 04-418Q), supplemented with 50 ng/mL mIL3, 50 ng/mL Flt3L, 50 ng/mL mTPO, and 100 ng/mL mSCF. A measure of 2 μg/mL Dox (Sigma, D9891) was used for MLL-AF9 oncogene induction. M-CSF and G-CSF were used at 20 ng/mL. Palbociclib (Selleckchem, S1579) dissolved in H2O was added to culture medium, and used at concentrations as indicated.

**Methylcellulose colony-forming assay**. GMPs were plated in MethoCult GF M3434 (Stem Cell Technologies) for colony forming, following manufacturer's instructions. For serial colony forming, colonies were scored after 7 days from initial plating, cells were collected, and replated as single-cell suspension.

For tracking colony formation at clonal level, micro-wells were casted using 1.2% low gelling temperature argarose (Sigma, A9045) with MicroTissues® 3D Petri Dish® (Sigma, Z764043). The micro-well gels were submerged in PBS for equilibration at 37 °C for 20 min, followed by x-vivo15 overnight. Medium was removed the next day, and GMPs resuspended at 2,500 cells/mL in 75 μl were loaded into each gel chamber, which were then incubated at 37 °C for 15 min to allow cells settling down into individual wells. Afterward, extra medium supplemented with cytokines was added to cover the entire gels in order to provide sufficient volume and nutrients for cell growth. After 2 days in liquid culture, medium was replaced with MethoCult GF M3434 for colony growing. Schema is shown in Supplementary Fig. 3a.

**Single GMP cell cycle rate density plot**. At each data point, a circle with radius 0.5 is used to determine area and data points included to calculate the local density. The color scale represents data density. The color scales of Fig. 2a, b and Fig. 8c are normalized to the same range, with the highest observed density set to 100.

**Image acquisition and processing**. For Giemsa images (Supplementary Fig. 2c), peripheral blood and bone marrow smears were stained with May-Grunwald Giemsa (Sigma MG500), and images were taken in bright field under 60× objective, using Olympus BX51TRF.

Colonies and micro-well images were acquired using ImageXpress Micro 4 high-content imaging system (Molecular Devices) at 4× or 10× objective. Images

from adjacent fields were then stitched together in ImageJ to generate the whole micro-well views. Colony size was quantified using MetaMorph image analysis software.

**GMP transplantation and induction of AML in vivo**. FACS-sorted GMPs were transplanted through tail vein into recipient mice that had received 6-Gy irradiation with gamma source. If oncogene was to be induced, 0.1 mg of Dox was given via intraperitoneal injection at the time of transplantation. Animals were then fed with 1 g/L Dox in drinking water sweetened with 10 g/L sucrose. Development of leukemia was monitored by periodic analysis of peripheral blood obtained by tail vein bleeding.

**Palbociclib treatment in vivo**. For in vivo palbociclib administration, palbociclib (Selleckchem S1579) was dissolved in PBS and given by intraperitoneal injection at 30 mg/kg body weight. Recipient mice were given three shots of palbociclib, 1-day prior to, the day of, and 1-day after the transplantation. Schema is shown in Supplementary Fig. 8h.

**Fluorouracil treatment in vivo**. To induce emergency myelopoiesis in vivo, fluorouracil (Mckesson #479124, 50 mg/mL) was given by intraperitoneal injection at 150 mg/kg body weight. GMPs were harvested from bone marrow 8 days post fluorouracil injection.

**RNA extraction, reverse transcription and QPCR**. Total RNA was extracted with TRIzol® Reagent (Ambion) and reverse transcribed using SuperScript™ III First-Strand Synthesis SuperMix (Invitrogen) following manufacturer's instructions. Quantitative real-time PCR was performed using the iQ™ SYBR® Green Supermix (Bio-Rad). Gene expression levels were normalized to GAPDH level in the same sample. Gene specific primers are listed in Supplementary Table 3.

**RNA-seq and data analysis**. The quality of total RNA was analyzed on Agilent Bioanalyzer. The RNA samples with more than eight RNA intergration number were chosen for RNA-seq library preparation using TruSeq Stranded mRNA Library Preparation Kit from Illumina (Cat # RS-122-2101). GMP samples treated with Dox for 24 h (Figs. 3, 5) were sequenced on HiSeq 2000 platform, and LKS/GMP/Mac1 + Gr1int samples treated with Dox for 2 days (Fig. 4) were sequenced on HiSeq 4000 platform as pair-end 100 cycles following manufacture's instruction. Sequencing reads were mapped to mouse mm10 using TopHat. Gene counts were quantified by featurecounts and DEGs were identified using DeSeq2, with adjusted $P < 0.05$. GO analysis of DEGs was performed through DAVID (https://david.ncifcrf.gov/). Scatter plots and heatmap were generated using ggplot2. Gene Set Enrichment Analysis (GSEA) was performed using GSEA software with default settings.

**ATAC-seq and data analysis**. ATAC libraries were prepared as previously described[52]. A total of 50,000 cells were used per reaction for each cell type. Libraries were sequenced on Illumina HiSeq 2500 platform at Yale Center for Genome Analysis (YCGA). Sequencing reads were trimmed for adapter sequences using Cutadapt and aligned to mouse mm10 using Bowtie2. Duplicates were removed using Picard from uniquely mapped reads, and MACS2 BAMPE mode was used to call peaks. Differential peaks were analyzed using Diffbind default settings, and visualizations were done using deepTools (version 2.5.7). GO analysis was performed using GREAT (http://great.stanford.edu/public/html/index.php).

**ScRNA-seq and data analysis**. GMPs harvested from mouse bone marrow were used either as fresh sample or cultured for one day +/−Dox. ScRNA-seq libraries were generated using the 10× Genomics Chromium Single Cell 3' Reagent Kit. Libraries were sequenced on Illumina HiSeq 2500 platform at YCGA as pair-end 75 bp cycles. The gene expression matrix was generated using CellRanger 2.1 with default parameters using the mm10 mouse genome indices from 10× genomics.

Single-cell libraries were processed in Python using the scprep package (github.com/krishnaswamylab/scprep). Cells with libraries smaller than 5500 UMI/cells were removed, and genes detected in fewer than 15 cells were removed. Gene expression data was library size normalized and square-root transformed. PHATE visualizations[25] were generated using default parameters. MELD and Vertex Frequency Clustering analysis[26] was run on +/−Dox samples using default parameters. For visualization and comparison of gene expression values between clusters, MAGIC was used to denoise and impute gene expression[53]. A small number of outlier cells (~1%) were found in the fresh GMPs that are discontinuous from the major population. The outliers were excluded from further analysis. Codes used in the analysis are as previously described[25,26,53].

**Patient data**. Patient data was downloaded from TARGET-AML (https://portal.gdc.cancer.gov/projects/TARGET-AML). Patients with associated transcriptome profiling data were analyzed for *CCND1* expressions and survival.

**Statistical analysis**. Paired or unpaired *t*-test was used in statistical analyses as specified in figure legends. *p* values were calculated based on two-tailed *t*-test. *F*-test was used to determine nonzero slope for linear regression, when calculating correlations. Nonparametric Kolmogorov–Smirnov test was used to calculate value for gene expressions (Fig. 3d). Log-rank (Mantel–Cox) test was used on Kaplan–Meier survival curves.

**Reporting Summary**. Further information on research design is available in the Nature Research Reporting Summary linked to this article.

## Data availability

RNA-seq, ATAC-seq, and single-cell 10× Genomics data that support the findings of this study have been deposited in Gene Expression Omnibus with the accession code GSE121768 [https://www.ncbi.nlm.nih.gov/geo/query/acc.cgi?acc = GSE121768].

Patient data was downloaded from TARGET-AML (https://portal.gdc.cancer.gov/projects/TARGET-AML).

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

## Acknowledgements
Research reported in this publication was supported by DP2GM123507 (S.G.), Charles Hood Foundation (S.G.), Gilead Sciences Research Scholars Program in Hematology/ Oncology award (S.G.) and F31HD097958 (D.B.). Studies related to normal hemato- poietic progenitors were also supported by U54DK106857 Yale Cooperative Hematology Specialized Core Center (YCCEH).

## Author contributions
X.C. designed, performed, analyzed most of the experiments, and wrote the manuscript. S.G. supervised the project, designed experiments, and wrote the manuscript. D.B. analyzed the scRNA-seq data. C.S. helped with 5FU mouse experiment. X.W. and M.Z. helped with RNA-seq. A.H., X.H., A.E., S.K. and S.G. provided intellectual support. All the authors reviewed the manuscript.

## Competing interests
The authors declare no competing interests.
