## [Peer Review File · Nature Communications]

Reviewers' comments:

Reviewer #1 (Remarks to the Author):

The cellular origin of acute myeloid leukemia (AML) is a subject of ongoing debates. Previous work has suggested that AML driven by some MLL fusion genes (e.g. MLL-AF9) most likely originates in the majority of the cases in granulocyte-macrophage progenitors (GMP). To better understand the underlying mechanisms the authors generated a transgenic DOX-regulated MLL-AF9 mouse line that allowed to follow transformation on a single cell/colony level (Fig. 1). They found that in vitro colony formation by GMP (used as read-out for transformation) was correlating to the number of cell divisions (Fig. 2). Comparative gene expression profiling and chromatin analysis suggested that MLL-AF9 early on preserved expression status of genes in fresh GMP (Fig. 3). Activation of MLL-AF9 in different hematopoietic stem and progenitor cells (LSK, GMP, Mac1+ myeloid cells) suggested that its transforming potential depends on the proliferative state of the starting cells (Fig.4). Finally, they show that impairing cell cycle progression in the initiating cells (e.g. by a small molecular CDK inhibitor) dampens the MLL-AF9 effect on gene expression (Fig.5) correlated with reduced colony formation in vitro and leukemia induction in vivo (Fig.6).

Overall the study is well-performed and the MS well-written with figures of high quality. In its current form the study is clearly very interesting for a very specialized readership. However, the study would clearly gain more importance and wider interest, if the authors could address whether the proposed mechanism is also valuable beyond the mouse model and important for the human disease.

Detailed comments:

1. What is the significance of these experimental findings for the human disease? It could be important to at least try to find any correlations of cycle-dependent transformation from the model (e.g. by comparing gene expression signatures) to MLL fusion-driven human AML?
2. Single cell/colony analysis showed that MLL-AF9 can only transform GMPs after 24h and 1-3 divisions (Fig. 2a-b) and the authors suggested that "insufficient amount of MLL-AF9 being induced at this time, or the induced oncogene not significantly altering the cellular state early on". These are valuable hypothesis that can be addressed by immunoblotting and ChIP-sequencing.
3. The author used a DOX-inducible MLL-fusion gene transgenic mouse line very similar to previous studies like Ugale et al. and Stavropoulou et al. (cited as references 9 & 13). The fusion was controlled from a TRE-minimal promoter in the Hprt locus on chromosome X indicating sex-dependent differences in expression levels? Did the authors use cells from female or male mice for their experiments? What about expression levels of the fusion: did the authors determine a dose-response curve (DOX vs. MLL-AF9 mRNA and/or protein) in male and female cells? What about leakiness of the system? Expression levels of the fusion seems to be critical for transformation as recently indicated by Liang et al. (Cell, 168:59-72, 2017) or Stavropoulou et al. (Hemasphere, 2:e51, 2018) and is most likely very important for the here performed experiments. Even though they used the IRES-driven Delta-NGFR marker as an expression readout, this will not provide information about MLL-AF9 protein expression. Was transformation in vitro and in vivo reversible upon removal of DOX?
4. They found higher expression of HoxA9 and Meis1 in freshly isolated LSK (=LT/ST/CMP) than GMP (20x, p-value?), however both HoxA9 and Meis1 was also consistently higher in GMP +DOX compared to those kept -DOX (6x, p-value?) in vitro (Fig.3). Even though the authors claimed that DOX/the fusion does not induce the classical stem-cell signature (due to low levels), the consistent and most likely significant difference in liquid medium (that normally does not favor expression of these genes that are more necessary for clonogenic growth) may also speak for the opposite!? The experiments here are classic examples how different culture conditions affect gene expression of hematopoietic cells: gene expression signatures of cells in liquid culture are often predominated by growth and survival genes, whereas those from cells in and semi-solid medium (e.g. MC) often

show additional marked expression of self-renewal genes. Freshly isolated BM cells are more similar to cells kept in semi-solid medium than expanded in a completely non-physiologic liquid culture. The authors have to be careful with conclusions when comparing different culture conditions.

5. Fig. 2a-b: It is not clear what the pseudo-colored normal cell number distribution measures? Is the 20-100 the percentage of cells in a given colony? Please clarify for the reader.

6. Fig. 2f-g: a different color/pattern could help to distinguish colonies prior and 24h after DOX addition.

Reviewer #2 (Remarks to the Author):

In the manuscript "MLL-AF9 Initiates Transformation from Fast-Proliferating Myeloid Progenitors", the authors demonstrate that a highly proliferating subpopulation of GMPs are the cell of origin for MLL mediated transformation. In a series of thoughtful and well executed experiments the authors work to establish that the proliferative rate is the critical determinant of transformation. However, the manuscript suffers from an outdated concept of GMP heterogeneity and differentiation hierarchy which complicates the interpretation of the data. The authors state that "An ideal experimental system to discern cellular traits underlying the permissiveness to transformation should satisfy the following criteria: (1) all cells are similar in developmental stage and oncogene expression, but only some transform to a malignant state;" however, the "GMP" does not truly satisfy this requirement of uniform developmental stage. Recent single cell transcriptional analysis of the GMP compartment has demonstrated that there exists heterogeneity within this population. Phenotypic markers can be used to identify "multi-lineage" GMPs (CD115-, Ly6C-), monocytic progenitors (CD115+, Ly6C+), and granulocytic progenitors (CD115-, Ly6C+) 1, 2. The interpretation that proliferation is critical for transformation is dependent on the assumption that the highly proliferative population does not correlate with one of these GMP subpopulations. At the same time, little is known about the proliferative rates and MLL-AF9 transformation potential of GMP subpopulations and addressing these questions would significantly improve the manuscript.

Major points

1. Determine the proliferative rates of the "multi-lineage", granulocytic, and monocytic GMPs using the established phenotypic markers. It has not yet been established whether any of the GMP subpopulations are more proliferative than the others and this would be a significant finding. If a single GMP population correlates with the highly proliferative subpopulation described in the manuscript then MLL-AFP transformation efficacy should be assessed.
2. The authors should perform scRNAseq to directly determine whether the transcriptional changes they describe in Fig 3 in response to MLL-AF9 induction are the result of changes actual transcriptional changes at the single cell level or rather changes in the composition of the GMP population. Alternatively, the authors could perform a comparative bioinformatic analysis with the dataset from Olsson et. Al. 2016.
3. The use of the CDK4/6 inhibitor strongly supports the theory that active proliferation is critical to transformation. However, it cannot be ruled out that the inhibitor has additional effects and a reciprocal experiment where the authors increases GMP proliferation prior to MLL transformation would greatly strengthen this point. The authors use a minimal set of cytokines that are unlikely to effectively support the proliferation of cells as they differentiate along the myeloid lineage. This is likely why colony forming ability is lost in DOX- culture prior to methyl-cellulose. Would addition of myelopoietic cytokines such as M-CSF, G-CSF, or IL-6 increase GMP proliferation and thereby increase MLL-AF9 transformation efficiency? Because the cytokines M-CSF and G-CSF are also instructive of lineage commitment³, this experiment would likely uncouple any relationship between differentiation and proliferation that might complicate interpretation of the previously presented results.

4. While there may not be such a tractable a model to measure initiation of transformation in human GMPs, it would at least be helpful to determine whether or not there exists a similar highly proliferative subpopulation.

REFERENCES

1. Olsson, A. et al. Single-cell analysis of mixed-lineage states leading to a binary cell fate choice. *Nature* 537, 698-702 (2016).
2. Yanez, A. et al. Granulocyte-Monocyte Progenitors and Monocyte-Dendritic Cell Progenitors Independently Produce Functionally Distinct Monocytes. *Immunity* 47, 890-902 e894 (2017).
3. Rieger, M.A., Hoppe, P.S., Smejkal, B.M., Eitelhuber, A.C. & Schroeder, T. Hematopoietic cytokines can instruct lineage choice. *Science* 325, 217-218 (2009).

Reviewer #3 (Remarks to the Author):

In this manuscript, Guo and colleagues attempted to test whether the heterogeneity of cell cycle rates in GMPs determines the probability of leukemia transformation by the MLL-AF9 fusion oncogene. They first established an in vitro culture and colony formation assay to evaluate the probability of MLL-AF9-mediated transformation starting from single hematopoietic cells. By counting cell numbers (as an indication of cell division) and subsequent colonies under various conditions (i.e. +/- Dox or +/- cell cycle inhibition by palbociclib), the authors observed an interesting correlation between fast cell cycle kinetics and permissiveness for transformation. Then by comparing gene expression and chromatin accessibility using RNA-seq and ATAC-seq analyses, the authors proposed a model that MLL-AF9 expression preserves the gene expression of the cellular states, and the rapidly-cycling more immature myeloid progenitors account for the cell-of-origin of MLL-AF9-mediated transformation.

Overall, this study utilizes an elegantly designed Dox-inducible MLL-AF9 knock-in allele and an in vitro culture model to examine the early steps of leukemia transformation at single cell level. The overall findings related to the correlation between cell cycle states and permissiveness for transformation is potentially interesting. However, there are important limitations with the in vitro assays that may select for a subset of GMP cells that better adapted the in vitro culture conditions. There are several important questions remain to be addressed, including how to distinguish between intrinsic cell cycling rate and MLL-AF9 oncogene expression levels in the transformability of different GMP cells, and how to distinguish whether the fast cycling behavior marks a unique cell lineage/state with inherent ability to transform instead of the high transformation probability itself. Therefore, despite the potentially interesting findings described in this study, the current manuscript suffers a number of weaknesses as outlined below that require significant work to make the study sound and convincing.

The specific comments:

1. The major concern is related to the in vitro cell culture model used in this study. In vitro culture of hematopoietic stem and progenitor cells may select for a small subset of cells that better adapted the culture conditions (medium, cytokines, etc). Thus it is unclear whether the observed correlation between fast cycling kinetics and high transformation rate was due to the in vitro selection process or cell intrinsic mechanisms. This limitation is noted by gene expression analysis that leukemia stem cell signature genes and MLL-AF9 targets (HoxA9 and Meis1) are significantly downregulated in cultured GMP-Dox compared to freshly isolated GMPs. MLL-AF9 expression partially restored the expression of HoxA9 and Meis1 (GMP +Dox; Fig. 3c). Similarly, the expression of many cell cycle related genes (Cdk6 and others) was altered by in vitro culture, and MLL-AF9 restored the gene expression of those genes to various extents. It is likely that the

observed correlation was due to selection of a subset of GMPs that proliferated better (therefore, with fast cycling kinetics) in vitro, allowing these cells to be transformed, whereas other cells did not adapt to the culture, failed to proliferate, and/or became differentiated. Similar limitations also apply to the comparison between the transformation potential of LKS, GMPs and Mac⁺ cells in vitro (Fig. 4). Therefore, it is critical to validate the findings that were based on in vitro culture studies using in vivo tracing studies to determine whether the correlation is present under physiological conditions.

2. It is also important to distinguish the effects from intrinsic cell cycling rate versus MLL-AF9 oncogene expression levels in the transformability of different GMP cells. The expression of MLL-AF9 at the single cell level should be measured in slow and fast cycling cells, as well as the cells that eventually become transformed.

3. As the authors discussed that the fast cycling behavior could simply mark the cell lineage/state with inherent ability to transform instead of the high transformation probability itself. In other words, it is important to determine whether the fast cycling behavior is the determining factor for transformation probability, or alternatively, the fast cycling behavior is part of the phenotypes associated with unique cell lineages/states (i.e. more immature myeloid progenitors) with high transformation potential. To test this, the authors assessed colony forming efficiency in GMPs with delayed Dox addition for 24 hours (Fig. 2f). They found a relatively stronger correlation between transformation potential and the cell cycle rate at the time of Dox treatment (24-48h) compared with prior to Dox treatment (0-24h). However, these experiments did not directly assess whether the fast cycling cells mark a distinct cell state that are associated with high transformation potential. Given these issues and to better assess the heterogeneity of GMPs, it is important to perform single cell RNA-seq analysis of fresh and cultured GMPs with or without MLL-AF9 expression, cluster cells based on cell cycle related gene signatures, and further examine the correlation with transformation potential in different cell clusters in vitro and in vivo.

4. The authors proposed that the rapidly-cycling more immature myeloid progenitors are the cell-of-origin of MLL-AF9-mediated transformation, however no evidence was provided to define the molecular and functional characteristics of these more immature progenitors. Thus scRNA-seq based studies may help better define the cell populations with high transformation potential.

5. The illustration for Fig. 2a,b should be improved, as currently presented, it is very difficult to understand and compare the differences between various conditions (0-24h vs 24-48h, -Dox vs +Dox, etc).

6. The authors used the top enriched GO terms for correlative analysis of changes in cell cycle, proliferation, or differentiation-related genes. GO type of analysis can often be affected by the number of differentially expressed genes, thus it is important to show expression of individual representative genes besides GO terms.

7. Some corroborative evidence is needed for the cell cycle analysis of palbociclib treatment to assess the extent to which the cell cycle was altered.

8. Statistical analysis is needed for several figure panels (i.e. Fig. 2, Fig. 3c-f, and Fig. 5b) since many of the observed effects was modest.

Point-by-point answers

Our answers are in this color.

Reviewer #1 (Remarks to the Author):

The cellular origin of acute myeloid leukemia (AML) is a subject of ongoing debates. Previous work has suggested that AML driven by some MLL fusion genes (e.g. MLL-AF9) most likely originates in the majority of the cases in granulocyte-macrophage progenitors (GMP). To better understand the underlying mechanisms the authors generated a transgenic DOX-regulated MLL-AF9 mouse line that allowed to follow transformation on a single cell/colony level (Fig. 1). They found that in vitro colony formation by GMP (used as read-out for transformation) was correlating to the number of cell divisions (Fig. 2). Comparative gene expression profiling and chromatin analysis suggested that MLL-AF9 early on preserved expression status of genes in fresh GMP (Fig. 3). Activation of MLL-AF9 in different hematopoietic stem and progenitor cells (LSK, GMP, Mac1+ myeloid cells) suggested that its transforming potential depends on the proliferative state of the starting cells (Fig.4). Finally, they show that impairing cell cycle progression in the initiating cells (e.g. by a small molecular CDK inhibitor) dampens the MLL-AF9 effect on gene expression (Fig.5) correlated with reduced colony formation in vitro and leukemia induction in vivo (Fig.6).

Overall the study is well-performed and the MS well-written with figures of high quality. In its current form the study is clearly very interesting for a very specialized readership. However, the study would clearly gain more importance and wider interest, if the authors could address whether the proposed mechanism is also valuable beyond the mouse model and important for the human disease.

We thank the reviewer for commending that our “study is well-performed and the MS well-written with figures of high quality”. We greatly appreciate your thoughtful comments and suggestions.

We have followed your suggestion to explore the relevance of our proposed mechanism in human AMLs. The new results, described in detail below and in the revised manuscript, suggest that our model in which the cell cycle rate of myeloid progenitors sets the probability of malignancy initiation, is likely applicable to human AMLs.

Detailed comments:

1. What is the significance of these experimental findings for the human disease? It could be important to at least try to find any correlations of cycle-dependent transformation from the model (e.g. by comparing gene expression signatures) to MLL fusion-driven human AML?

To examine the relevance of our findings in human AML, we have performed new analyses of publicly available datasets. Specifically, we have analyzed the expression of cell cycle genes in human GMPs as described by (Buenrostro et al., 2018). The results suggest that human GMP-A subset express higher levels of *CDK6* and *HOXA9* than the other two closely related subsets GMP-B and GMP-C. GMP-A is also the only subset with detectable level of *CCND1* (new Fig. 9a). In the TARGET-AML patient cohort (Farrar et al., 2016), we further identified that *CCND1* expression stratifies MLL-patient survival, with the high expression cases having significantly worse prognosis (new Fig. 9b). In contrast, *CCND1* level does not have prognostic value in non-

MLL AMLs (new Fig. 9c). These results are described as the last paragraph of the main text and suggest that human AML leukemogenesis by MLL-fusion oncogenes is likely also influenced by cell cycle rate of the initiating cells.

2. Single cell/colony analysis showed that MLL-AF9 can only transform GMPs after 24h and 1-3 divisions (Fig. 2a-b) and the authors suggested that “insufficient amount of MLL-AF9 being induced at this time, or the induced oncogene not significantly altering the cellular state early on”. These are a valuable hypothesis that can be addressed by immunoblotting and ChIP-sequencing.

To attempt immunoblotting and ChIP for MLL-AF9, we have tested two independent commercial antibodies against human MLL1 which shares identical N-terminal half as our iMLL-AF9 transgenic oncoprotein: Bethyl Laboratories catalog # A300-086A and Abcam catalog # Abcam ab32400. Neither antibody produced convincing band on western blots even in the positive controls (whole cell extracts from MLL-AF9 transformed leukemia cells, data not shown). The poor performances of these antibodies on western blots dampened our confidence in obtaining convincing results by ChIP when the number of primary GMPs are low. This technical challenge is partly reflected by previous publications where MLL-AF9 ChIP was performed by indirect approaches (e.g. the widely cited paper by (Bernt et al., 2011), used biotinylated MLL-AF9).

Due to the concerns with antibody quality against MLL-AF9 itself, we measured instead Δ NGFR expression encoded by the same transgenic cassette by flow cytometry. Our analysis indicated that Δ NGFR levels reached plateau after 12 hours of Dox induction (data not shown in manuscript but included here), suggesting that the transgene cassette should have been sufficiently induced by 24 hours. We therefore have revised our text to state:

“The lack of difference in cycling rates during the first 24-hours of Dox treatment offers a brief time window to capture the intrinsic GMP cell cycle rate, before it had been significantly altered by the oncogene”.

3. The author used a DOX-inducible MLL-fusion gene transgenic mouse line very similar to previous studies like Ugale et al. and Stavropoulou et al. (cited as references 9 & 13). The fusion was controlled from a TRE-minimal promoter in the Hprt locus on chromosome X indicating sex-dependent differences in expression levels? Did the authors use cells from female or male mice for their experiments? What about expression levels of the fusion: did the authors determine a dose-response curve (DOX vs. MLL-AF9 mRNA and/or protein) in male and female cells?

Although homozygous knock-in female cells carry two copies of the iMLL-AF9 allele, only one of these should be active due to dosage compensation by X chromosome inactivation. Transgene expression level plateaus at 2µg/ml Dox, a concentration we have used throughout the manuscript. We have experimentally confirmed these points, and added the following text with accompanying figures/legends.

“As the iMLL-AF9 allele was targeted onto the X chromosome, which differs in copy number between male and female animals, we first compared transgene inducibility in hematopoietic progenitors of both sexes. As expected from X chromosome inactivation in female cells, GMPs from homozygous iMF9/iMF9 females showed similar Dox-dependent transgene induction as those isolated from iMF9/Y males, when both sexes were homozygous for the rtTA allele (*r/r*) (Extended Data Fig. 1a-b). Dox-treated cells from both sexes also displayed comparable expression of key MLL-AF9 target genes such as *Hoxa9* (Extended Data Fig. 1c). Thus, all experiments were performed using homozygous females or males for the iMLL-AF9 allele and homozygous rtTA.”

What about leakiness of the system? Expression levels of the fusion seems to be critical for transformation as recently indicated by Liang et al. (Cell, 168:59-72, 2017) or Stavropoulou et al. (Hemisphere, 2:e51, 2018) and is most likely very important for the here performed experiments. Even though they used the IRES-driven Delta-NGFR marker as an expression readout, this will not provide information about MLL-AF9 protein expression.

To attempt immunoblotting for MLL-AF9, we have tested two independent commercial antibodies against human MLL1 which shares identical N-terminal half as our iMLL-AF9 transgenic oncoprotein: Bethyl Laboratories catalog # A300-086A and Abcam catalog # Abcam ab32400. Neither antibody produced convincing band on western blots even in the positive controls (whole cell extracts from MLL-AF9 transformed leukemia cells, data not shown). Due to this technical reason, we did not further pursue examining MLL-AF9 protein expression. We have however obtained results from a number of complimentary and independent assays, which suggest that leaky oncoprotein expression should be insignificant, detailed below.

- 1) In the absence of Dox, MLL-AF9 was not detectable by RT-Q-PCR (Extended Data Fig. 1b, Extended Data Fig. 10e).
- 2) Cell cycle rates of wild type GMPs and iMLL-AF9 GMPs in the absence of Dox were identical (Fig 8a. identical to Extended Data Fig. 10d).
- 3) No serial replating colony could form in the absence of Dox (Fig. 1b).
- 4) Mice transplanted with iMLL-GMPs but not fed with Dox water never develop AML (Fig. 1c). All iMLL-AF9 mice remain healthy on regular water (data not shown).

From comparisons made on transgene expression, cell cycle analysis and transformed phenotypes, the above analyses collectively support that the phenotype we observed should not have been contributed significantly by leakiness.

Was transformation *in vitro* and *in vivo* reversible upon removal of DOX?

Transformation *in vitro* depends on sustained expression of MLL-AF9, as Dox removal led to reduced proliferation and colony formation (Extended Data Fig. 1d-f). We have added the following text with accompanying figures/legends.

“Transformation was dependent on sustained MLL-AF9 expression, as Dox removal greatly decreased proliferation in liquid cultures accompanied with reduced colony-formation in methylcellulose (Extended Data Fig. 1d-f), consistent with a similarly targeted MLL-ENL model (Stavropoulou et al., 2018).”

The phenotype/pathology following Dox removal *in vivo* could be complicated by secondary tissue changes. For example, we have reported pervasive multi-tissue fibrosis following Dox-driven myeloproliferation, revealed upon Dox removal (Guo et al., 2012). Given this potential complication, and that our central focus is the initiation of transformation, we did not further attempt the *in vivo* Dox removal experiment.

4. They found higher expression of HoxA9 and Meis1 in freshly isolated LSK (=LT/ST/CMP) than GMP (20x, p-value?), however both HoxA9 and Meis1 was also consistently higher in GMP +DOX compared to those kept -DOX (6x, p-value?) *in vitro* (Fig.3). Even though the authors claimed that DOX/the fusion does not induce the classical stem-cell signature (due to low levels), the consistent and most likely significant difference in liquid medium (that normally does not favor expression of these genes that are more necessary for clonogenic growth) may also speak for the opposite!?! The experiments here are classic examples how different culture conditions affect gene expression of hematopoietic cells: gene expression signatures of cells in liquid culture are often predominated by growth and survival genes, whereas those from cells in and semi-solid medium (e.g. MC) often show additional marked expression of self-renewal genes. Freshly isolated BM cells are more similar to cells kept in semi-solid medium than expanded in a completely non-physiologic liquid culture. The authors have to be careful with conclusions when comparing different culture conditions.

We thank the reviewer for this important cautionary note.

We have clarified the rationale for performing gene expression with cultured GMPs, by adding

“... these results indicate that the changes in cellular states during the brief culture renders GMPs to forfeit their colony-forming potential, which is preserved by the induced MLL-AF9 during this time. These results suggest that the molecular changes occurred during the brief culture could help to define the cellular states from which MLL-AF9 can initiate transformation *de novo*.”

Additionally, we have more clearly acknowledged the context in which the data is being interpreted, to state:

“This initial gene expression change is consistent with a lack of extensive reactivation of the stem cell program, at least when MLL-AF9 is induced during this brief culture.”

The layout of the accompanying Fig. 3a has also been adjusted to better indicate the relevant comparisons.

5. Fig. 2a-b: It is not clear what the pseudo-colored normal cell number distribution measures? Is the 20-100 the percentage of cells in a given colony? Please clarify for the reader.

We apologize for the confusion and have revised the figure and text for better clarity.

Specifically, the plots shown in Fig. 2a-b are similar to a flow cytometry plot, where the pseudo-color denotes population density. Higher values (redder in color) indicate more GMP lineages in a

specific region of the plot. The revised figures have contour lines to better indicate this feature. We have also revised the figure legend to the following:

“Cell cycle rates of individual iMLL-AF9 GMP lineages as determined by the number of divisions during the first 24-hours (x-axis) or the second 24-hours (y-axis) of culture. Dox was added at 0 hour, when the presence of single GMP was confirmed. Each dot denotes an individual GMP lineage. (a) –Dox: n=268; (b) +Dox: n=378. The number of divisions during the first 24-hour = $\text{Log}_2(\text{cell number at T24h} / \text{cell number at T0h})$; the number of divisions during the second 24-hour = $\text{Log}_2(\text{cell number at T48h} / \text{cell number at T24h})$. Pseudo color denotes GMP lineage density. Dotted diagonal line denotes $y=x$, indicating comparable cell cycle rates during the first and second 24-hours of culture. In the presence of Dox, significant number of GMP lineages underwent cell cycle acceleration, shown as the dense population above the diagonal line.”

6. Fig. 2f-g: a different color/pattern could help to distinguish colonies prior and 24h after DOX addition.

We thank the reviewer for this nice suggestion. We have revised Fig. 2f-g with orange bar denoting the time prior to Dox addition, and red bar denoting the 24-hours after Dox addition.

Reviewer #2 (Remarks to the Author):

In the manuscript “MLL-AF9 Initiates Transformation from Fast-Proliferating Myeloid Progenitors”, the authors demonstrate that a highly proliferating subpopulation of GMPs are the cell of origin for MLL mediated transformation. In a series of thoughtful and well executed experiments the authors work to establish that the proliferative rate is the critical determinant of transformation. However, the manuscript suffers from an outdated concept of GMP heterogeneity and differentiation hierarchy which complicates the interpretation of the data. The authors state that “An ideal experimental system to discern cellular traits underlying the permissiveness to transformation should satisfy the following criteria: (1) all cells are similar in developmental stage and oncogene expression, but only some transform to a malignant state;”, however, the “GMP” does not truly satisfy this requirement of uniform developmental stage. Recent single cell transcriptional analysis of the GMP compartment has demonstrated that there exists heterogeneity within this population. Phenotypic markers can be used to identify “multi-lineage” GMPs (CD115-, Ly6C-), monocytic progenitors (CD115+, Ly6C+), and granulocytic progenitors (CD115-, Ly6C+) 1, 2. The interpretation that proliferation is critical for transformation is dependent on the assumption that the highly proliferative population does not correlate with one of these GMP subpopulations. At the same time, little is known about the proliferative rates and MLL-AF9 transformation potential of GMP subpopulations and addressing these questions would significantly improve the manuscript.

We thank the reviewer for commending our work to be “thoughtful and well executed”. We greatly appreciate your thoughtful comments and suggestions.

Considering the heterogeneity among GMPs, we have removed the text stating “An ideal experimental system...” as you pointed out above.

We have performed new experiments and data analyses to investigate the heterogeneity among GMPs and added new sections on comparing the gene expression, cell cycle rate and transformation potential of the “multi-lineage” GMPs (CD115⁻, Ly6C⁻), monocytic progenitors (CD115⁺, Ly6C⁺) and granulocytic progenitors (CD115⁻, Ly6C⁺). These are presented as new Fig. 4, 8 and Extended Data Fig. 5,10, detailed below.

Major points

1. Determine the proliferative rates of the “multi-lineage”, granulocytic, and monocytic GMPs using the established phenotypic markers. It has not yet been established whether any of the GMP subpopulations are more proliferative than the others and this would be a significant finding. If a single GMP population correlates with the highly proliferative subpopulation described in the manuscript then MLL-AFP transformation efficacy should be assessed.

We have performed new experiments as suggested, and report these results in new Fig. 8 and Extended Data Fig. 10.

We have flow sorted GMPs into three subsets: multi-lineage progenitors (Ly6C⁻ CD115^{lo}), granulocytic progenitors (Ly6C⁺ CD115^{lo}), and monocytic progenitors (Ly6C⁺ CD115^{hi}) (Extended Data Fig. 10a). Using our microwell proliferation assay, we have determined the cell cycle rates of all three FACS-fractionated subpopulations. The results show that similar cell cycle rate heterogeneity remains in each subset (Fig. 8, Extended Data Fig. 10).

We have determined the transformation efficiency of these three subsets. The results show that transformation efficiency correlated with cell cycle rate within each cell subset (Fig. 8h). These data also show that a faster dividing more differentiated cell could be more permissive to transformation than its slower dividing multipotent predecessor.

These results are described in a new section titled “Molecular features of the cells initiating MLL-fusion leukemia *in vivo*”.

2. The authors should perform scRNAseq to directly determine whether the transcriptional changes they describe in Fig 3 in response to MLL-AF9 induction are the result of changes actual transcriptional changes at the single cell level or rather changes in the composition of the GMP population. Alternatively, the authors could perform a comparative bioinformatic analysis with the dataset from Olsson et al. 2016.

We have performed scRNA-seq of iMLL-AF9 GMPs and the results are shown as new Fig. 4 and Extended Data Fig. 5. A new section describing these results have been added under the title “Single cell RNA-seq reveals a subset of GMPs that express high levels of cell cycle genes and MLL-AF9 target genes independent of oncogene exposure”. We have also compared our data with those by Olsson et al., 2016 in multiple context. These new experiments/analyses revealed the following insights:

- 1) Gene expressions of single GMPs cultured for one day vary substantially, irrespective of Dox treatment. This heterogeneity is reflected by their distinct expression level of MLL-AF9 target genes (Fig. 4a), cell cycle genes (Extended Data Fig. 5a), and differentiation genes (Extended Data Fig. 5b). These results support the existence/extent of cell state heterogeneity among GMPs.

- 2) The overall population structure, based on overall gene expression, between the GMPs cultured with or without Dox largely overlap (Fig. 4b). Dox treatment enriched a subpopulation occupying a specific position on the overall trajectory (Fig. 4c).
- 3) In agreement with the gene expression of GMPs by Olsson et al., all cultured GMPs could be clustered into five subsets (Fig. 4d-e): Clusters 1 and 5 express low levels of both *Ly6c2* and *Csf1r*, cluster 3 is distinguished by the presence of *Ly6c2* and absence of *Csf1r*, and clusters 2 and 4 express both *Ly6c2* and *Csf1r* (Extended Data Fig. 5e). Clusters 1 and 5 resemble the least mature subset due to their low expression of *Ly6c2* and *Csf1r*, which also express higher *Cdk4/6* (Fig. 4e, Extended Data Fig. 5e,f).
- 4) Importantly, within each cluster, the expression levels of *HoxA9/Meis1/Mef2c* in +Dox GMPs were present at similar levels as in -Dox GMPs, though population frequencies changed (Fig. 4g). Specifically, Cluster 1 & 5 increased in the presence of Dox.

Overall, these results revealed that a subset of GMPs expressing high levels of *Cdk4/6* and *Hoxa9/Meis1/Mef2c* exist independent of Dox treatment. The immediate genomic consequence to MLL-AF9 expression in individual cells is consistent with preservation of the already existing gene expression programs in single cells, such that the overall population structure remains largely unchanged at this time. Therefore, the enrichment of cells bearing such gene expression in the presence of Dox should have contributed the gene expression differences seen by bulk RNA-seq (Fig. 3c).

3. The use of the CDK4/6 inhibitor strongly supports the theory that active proliferation is critical to transformation. However, it cannot be ruled out that the inhibitor has additional effects and a reciprocal experiment where the authors increases GMP proliferation prior to MLL transformation would greatly strengthen this point.

We have used an emergency myelopoiesis model (induced by 5FU injection) to increase GMP cell cycle, since the further activation of GMP cell cycle has been well-defined temporally and quantitatively by (Herault et al., 2017). The results of these new experiments are presented as new Fig. 7 and Extended Data Fig. 9. The main findings are described in two new paragraphs under the section "Efficiency of MLL-AF9-mediated leukemogenesis responds to cell cycle modulation of the initiating GMPs".

- 1) In this model, enhanced GMP proliferation occurs 8-days post 5FU injection (Herault et al., 2017), detectable by EdU pulse labeling (Extended Data Fig 9a). Under this condition, our microwell cell cycle analysis revealed increased frequency of GMPs capable of dividing three or more times within 24-hours (Fig. 7a).
- 2) We have determined the efficiency of transformed colony formation by these regenerating GMPs. The efficiency of forming transformed colonies again correlated with cell cycle rates (Fig. 7c), similar to the behavior of homeostatic iMLL-AF9 GMPs (Fig. 2e).
- 3) 5FU-activated iMLL-AF9 GMPs induced leukemogenesis with shortened latency (Fig. 7d-f), confirming increased transformation by these regenerating GMPs *in vivo*.

These new data demonstrate that MLL-AF9 mediated transformation is potentiated by GMP cell cycle acceleration, complementing the results by the CDK4/6 inhibitor.

The authors use a minimal set of cytokines that are unlikely to effectively support the proliferation of cells as they differentiate along the myeloid lineage. This is likely why colony

forming ability is lost in DOX- culture prior to methyl-cellulose. Would addition of myelopoietic cytokines such as M-CSF, G-CSF, or IL-6 increase GMP proliferation and thereby increase MLL-AF9 transformation efficiency? Because the cytokines M-CSF and G-CSF are also instructive of lineage commitment³, this experiment would likely uncouple any relationship between differentiation and proliferation that might complicate interpretation of the previously presented results.

To test whether limited growth factor/cytokine is responsible for the lost transformation potential during culture, we added additional cytokines as the reviewer suggested. Addition of M-CSF, G-CSF, or IL6 did not significantly change the cell cycle rate of GMPs during the two-day culture (new Extended Data Fig. 3e), suggesting that the original cytokine cocktail was sufficient. The reduction in colony formation following the two-day liquid culture (Extended Data Fig. 3d) was not rescued by added M-CSF, G-CSF, and IL6 (new Extended Data Fig. 3f). These results are accompanied by text stating:

“The decrease in colony formation by these briefly cultured GMPs is consistent with their partial differentiation, and not due to insufficient amount of growth factors/cytokines during the two-day culture, as addition of IL6, M-CSF, and G-CSF did not increase their proliferation rate within this time window (Extended Data Fig. 3e) or alleviate the decrease in colony formation following the two-day culture (Extended Data Fig. 3f).”

4. While there may not be such a tractable a model to measure initiation of transformation in human GMPs, it would at least be helpful to determine whether or not there exists a similar highly proliferative subpopulation.

Thank you for this thoughtful suggestion.

To examine the relevance of our findings in human GMPs and AML, we have analyzed the expression of cell cycle genes in human GMPs as described by (Buenrostro et al., 2018). The results suggest that human GMP-A subset express higher levels of *CDK6* and *HOXA9* than the other two closely related subsets GMP-B and GMP-C. GMP-A is also the only subset with detectable level of *CCND1* (new Fig. 9a). We further extended our analyses to the TARGET-AML patient cohort (Farrar et al., 2016), and identified that *CCND1* expression stratifies MLL-patient survival, with the high expression cases having significantly worse prognosis (new Fig. 9b). In contrast, *CCND1* level does not have prognostic value in non-MLL AMLs (new Fig. 9c). These results are described as the last paragraph of the main text and suggest that human AML leukemogenesis by MLL-fusion oncogenes is likely also influenced by cell cycle rate of the initiating cells, e.g. the GMP-A.

REFERENCES

1. Olsson, A. et al. Single-cell analysis of mixed-lineage states leading to a binary cell fate choice. *Nature* 537, 698-702 (2016).
2. Yanez, A. et al. Granulocyte-Monocyte Progenitors and Monocyte-Dendritic Cell Progenitors Independently Produce Functionally Distinct Monocytes. *Immunity* 47, 890-902 e894 (2017).
3. Rieger, M.A., Hoppe, P.S., Smejkal, B.M., Eitelhuber, A.C. & Schroeder, T. Hematopoietic cytokines can instruct lineage choice. *Science* 325, 217-218 (2009).

Reviewer #3 (Remarks to the Author):

In this manuscript, Guo and colleagues attempted to test whether the heterogeneity of cell cycle rates in GMPs determines the probability of leukemia transformation by the MLL-AF9 fusion oncogene. They first established an in vitro culture and colony formation assay to evaluate the probability of MLL-AF9-mediated transformation starting from single hematopoietic cells. By counting cell numbers (as an indication of cell division) and subsequent colonies under various conditions (i.e. +/- Dox or +/- cell cycle inhibition by palbociclib), the authors observed an interesting correlation between fast cell cycle kinetics and permissiveness for transformation. Then by comparing gene expression and chromatin accessibility using RNA-seq and ATAC-seq analyses, the authors proposed a model that MLL-AF9 expression preserves the gene expression of the cellular states, and the rapidly-cycling more immature myeloid progenitors account for the cell-of-origin of MLL-AF9-mediated transformation.

Overall, this study utilizes an elegantly designed Dox-inducible MLL-AF9 knock-in allele and an in vitro culture model to examine the early steps of leukemia transformation at single cell level. The overall findings related to the correlation between cell cycle states and permissiveness for transformation is potentially interesting. However, there are important limitations with the in vitro assays that may select for a subset of GMP cells that better adapted the in vitro culture conditions. There are several important questions remain to be addressed, including how to distinguish between intrinsic cell cycling rate and MLL-AF9 oncogene expression levels in the transformability of different GMP cells, and how to distinguish whether the fast cycling behavior marks a unique cell lineage/state with inherent ability to transform instead of the high transformation probability itself. Therefore, despite the potentially interesting findings described in this study, the current manuscript suffers a number of weaknesses as outlined below that require significant work to make the study sound and convincing.

We thank the reviewer for commending our study that “utilizes an elegantly designed Dox-inducible MLL-AF9 knock-in allele and an in vitro culture model to examine the early steps of leukemia transformation at single cell level.” We greatly appreciate your thoughtful comments and suggestions.

We have performed a substantial amount of new experiments and data analyses to resolve your questions. As a result, a large number of new figures (Fig. 4,7,8,9 and Extended Data Fig. 1,5,9,10) and the accompanying texts have been added, detailed below.

The specific comments:

1. The major concern is related to the in vitro cell culture model used in this study. In vitro culture of hematopoietic stem and progenitor cells may select for a small subset of cells that better adapted the culture conditions (medium, cytokines, etc). Thus it is unclear whether the observed correlation between fast cycling kinetics and high transformation rate was due to the in vitro selection process or cell intrinsic mechanisms. This limitation is noted by gene expression analysis that leukemia stem cell signature genes and MLL-AF9 targets (HoxA9 and Meis1) are significantly downregulated in cultured GMP-Dox compared to freshly isolated GMPs. MLL-AF9 expression partially restored the expression of HoxA9 and Meis1 (GMP +Dox; Fig. 3c). Similarly, the expression of many cell cycle related genes (Cdk6 and others) was altered by in vitro culture, and MLL-AF9 restored the gene expression of those genes to various extents. It is likely that the observed correlation was due to selection of a subset of GMPs that proliferated better (therefore, with fast cycling kinetics) in vitro, allowing these cells to be transformed,

whereas other cells did not adapt to the culture, failed to proliferate, and/or became differentiated. Similar limitations also apply to the comparison between the transformation potential of LKS, GMPs and Mac+ cells *in vitro* (Fig. 4). Therefore, it is critical to validate the findings that were based on *in vitro* culture studies using *in vivo* tracing studies to determine whether the correlation is present under physiological conditions.

The choice of measuring cell cycle rate during the brief *in vitro* culture was necessitated by the unparalleled resolution/accessibility it could offer. Our attempt to address your concerns are two-fold.

First, we point to evidences supporting that the observed cell cycle rates *in vitro*, and its association with transformation efficiency, could not be simply accounted for by the culture condition.

- 1) Leukemogenesis by GMPs is responsive to their cell cycle manipulations *in vivo*. Specifically, when recipients were transiently treated with palbociclib after transplantation of the iMLL-AF9 GMPs, leukemogenesis was reduced (Fig 6e-h). Conversely, the regenerating GMPs during emergency myelopoiesis gave rise to more aggressive leukemia *in vivo* (new Fig 7d-f). In both instances, changes in leukemogenesis *in vivo* agreed with the respective changes in gene expression and colony formation *in vitro* (Fig 6,7). One example to be noted here is that the 5FU-activated GMP cell cycle, as describe by (Herault et al., 2017), was readily detected by our *in vitro* cell cycle rate assay (Fig. 7a), as well as by higher EdU+% following pulse labeling (new Extended Data Fig. 9a). These results demonstrate that the cell rate change detected by our *in vitro* assay is corroborated by independent assays using cells directly taken from *in vivo*.
- 2) We have performed new experiments to show that addition of M-CSF, G-CSF, or IL6 did not significantly change the cell cycle rate of GMPs during the two-day culture (new Extended Data Fig. 3e). The reduction in colony formation following the two-day culture (Extended Data Fig. 3d) was not rescued either by these extra growth factors/cytokines (new Extended Data Fig. 3f). These data demonstrate that changing culture conditions by adding more growth factors did not significantly affect the cell cycle rate during the time window tested. Furthermore, the cell cycle rate of -Dox GMPs (most divided 0 to 3 times within a 24-hour time widow, Fig. 2a), is highly consistent with our previous measurement of these cells cultured in a completely different condition, optimized for mouse embryonic stem cells (Guo et al., 2014). The highly consistent GMP cell cycle rate across three rather distinct culture conditions argue against the possibility that the fast cell cycle rate detected by our assay was specific to the culture conditions.
- 3) If faster-proliferating cells were better selected by culture, they should likely have been also better at forming colonies in the absence of Dox. This was not the case, as shown in Fig. 2e (grey bars). Similarly, if high cell cycle rate merely marks a cell/lineage with better survival/adaptation, then the faster-proliferating cells before the delayed Dox addition should likely have been more efficient to transform. However, cell cycle rates prior to Dox addition had insignificant correlation (Fig. 2g-h). These data argue against the possibility that the reduced colony forming ability associated with *in vitro* culture is mainly afforded by selection/adaptation.

Second, given the necessity to perform part of the studies *in vitro*, we have more clearly acknowledged the context in which the data is being interpreted, to state:

“...This initial gene expression change is consistent with a lack of extensive reactivation of the stem cell program, at least when MLL-AF9 is induced during this brief culture.”

following the clarified rationale for performing parallel gene expression analyses in cultured GMPs, by adding

“... these results indicate that the changes in cellular states during the brief culture renders GMPs to forfeit their colony-forming potential, which is preserved by the induced MLL-AF9 during this time. These results suggest that the molecular changes occurred during the brief culture could help to define the cellular states from which MLL-AF9 can initiate transformation *de novo*.”

2. It is also important to distinguish the effects from intrinsic cell cycling rate versus MLL-AF9 oncogene expression levels in the transformability of different GMP cells. The expression of MLL-AF9 at the single cell level should be measured in slow and fast cycling cells, as well as the cells that eventually become transformed.

We addressed this question by performing a series of new experiments/analyses via multiple approaches.

- 1) We performed scRNA-seq in iMLL-AF9 GMPs +/- Dox (new Fig. 4). The mRNA of the oncogene was too low to be detected in most cells. To overcome this problem, we used the mRNA level of Δ NGFR, which can be readily detected in many cells, as a proxy to estimate transgene level variation. We categorized cells into transgene-high and transgene-low cells, and compared their expression of cell cycle genes and transformation related genes (new Extended Data Fig. 5c). The results show that regardless of the transgene expression levels, cell cycle and transformation related genes were expressed at similar levels. These results are consistent with the notion that the expression of transformation-related genes is not significantly biased by the oncogene level, and/or that the level of oncogene expression is largely indistinguishable across single cells induced *in vitro*.
- 2) To complement the mRNA analysis in single cells, and to directly compare oncogene induction levels, we performed RT-Q-PCR of MLL-AF9 mRNA in three GMP subsets, prospectively isolated based on their cell surface expression of Ly6C and CD115 (new Extended Data Fig. 10a). These subsets differ in differentiation stage (Yanez et al., 2015) and cell cycle gene expression (new Fig. 8e,f). Importantly, in all three subsets, MLL-AF9 mRNA was not detectible without Dox but became highly induced by Dox (new Extended Data Fig. 10e). The minor differences in the level of induced MLL-AF9 mRNA among the subsets did not coincide with their permissiveness to transformation (new Fig. 8h). In contrast, transformation efficiency correlated with cell cycle rate within each cell subset and a faster dividing more differentiated cell could be more permissive to transformation than its slower dividing multipotent predecessor.
- 3) To assess transgene protein expression in single cells, we measured the co-transcribed/co-translated Δ NGFR levels by FACS in GMPs following a range of Dox concentrations (new Extended Data Fig. 1a). With all tested Dox concentrations, induced GMPs displayed a narrow unimodal peak, indicating the lack of major difference in transgene expression across many cells.

These multiple lines of evidence support that the observed cell cycle rate heterogeneity should not have been primarily contributed by oncogene expression level.

3. As the authors discussed that the fast cycling behavior could simply mark the cell lineage/state with inherent ability to transform instead of the high transformation probability itself. In other words, it is important to determine whether the fast cycling behavior is the determining factor for transformation probability, or alternatively, the fast cycling behavior is part of the phenotypes associated with unique cell lineages/states (i.e. more immature myeloid progenitors) with high transformation potential. To test this, the authors assessed colony forming efficiency in GMPs with delayed Dox addition for 24 hours (Fig. 2f). They found a relatively stronger correlation between transformation potential and the cell cycle rate at the time of Dox treatment (24-48h) compared with prior to Dox treatment (0-24h). However, these experiments did not directly assess whether the fast cycling cells mark a distinct cell state that are associated with high transformation potential. Given these issues and to better assess the heterogeneity of GMPs, it is important to perform single cell RNA-seq analysis of fresh and cultured GMPs with or without MLL-AF9 expression, cluster cells based on cell cycle related gene signatures, and further examine the correlation with transformation potential in different cell clusters *in vitro* and *in vivo*.

Thank you for this thoughtful suggestion. We have approached this question on multiple levels, which collectively support that cell states capable of fast cell cycle have high transformation potential.

- 1) We have performed single cell RNA-seq on fresh and cultured GMPs +/-Dox (Fig. 4), and clustered cells based on their overall gene expression similarity using PHATE algorithm (Moon et al., 2018, bioRxiv) and MELD (Burkhardt et al., 2019, bioRxiv), described in the revised manuscript in detail. GMPs cultured with or without Dox exhibited two main diverging trajectories, likely corresponding to the granulocyte- or monocyte-committed progenitors respectively, as determined by their expression of *Ly6c2* and *Csf1r* (Extended Data Fig. 5e). Our analyses yielded five clusters (Fig.4b-g): Clusters 1 & 5 expressed the lowest level of *Ly6c2* and *Csf1r* (Extended Data Fig. 5e), but highest *Cdk4/6* (Fig 4f) and *HoxA9/Meis1/Mef2c* independent of Dox (Fig. 4g). These results demonstrate that existence of cell states with high *Cdk4/6* expression does not require the oncogene, but became enriched with oncogene expression. Molecularly, the *Ly6c2^{Lo}/Csf1^{Lo}* cells resemble the previously reported metastable GMPs by (Olsson et al., 2016) by their low expression of *Irf8* and *Cebpe* (Fig. 8f).
- 2) Furthermore, we fractionated GMPs into three subpopulations based on cell surface Ly6C and CD115 expressions (new Fig. 8): multi-lineage progenitors (Ly6C- CD115^{lo}), granulocytic progenitors (Ly6C+ CD115^{lo}), and monocytic progenitors (Ly6C+ CD115^{hi}). These three subsets express different levels of *Cdk4/6*, with the multi-lineage progenitors (Ly6C- CD115^{lo}) expressing the highest amount (Fig. 8e). Using our micro-well proliferation assay, we revealed that similar cell cycle rate heterogeneity remains in each subset (Fig 8a-d). Within each subset, transformation efficiency remains correlated with cell cycle rate (Fig. 8h). Importantly, a faster dividing more differentiated cell could be more permissive to transformation than its slower dividing multipotent predecessor (Fig. 8h). A large portion of the Ly6C- CD115^{lo} multi-lineage progenitors could undergo further cell cycle acceleration during culture, consistent with their high *Cdk4/6* expression (Fig 8e). These data further support that the cell state efficient for transformation is distinguished by fast cell cycle rate.
- 3) Functionally, leukemogenesis by GMPs is responsive to their cell cycle manipulations *in vivo*. Specifically, when recipients were transiently treated with palbociclib when

transplanted with the iMLL-AF9 GMPs, leukemogenesis was reduced (Fig 6e-h). Conversely, GMPs during emergency myelopoiesis gave rise to more aggressive leukemia *in vivo* (new Fig 7d-f). Therefore, cell cycle modulation in both instances changed leukemogenesis *in vivo* (Fig 6,7).

- 4) To further explore the relationship between proliferative cell states and transformation, we have analyzed the expression of cell cycle genes in human GMPs as described by (Buenrostro et al., 2018). We extended our analyses to the TARGET-AML patient cohort (Farrar et al., 2016), and identified that *CCND1* expression stratifies MLL-patient survival, with the high expression cases having significantly worse prognosis (Fig. 9b). In contrast, *CCND1* level does not have prognostic value in non-MLL AMLs (Fig. 9c). These results are described as the last paragraph of the main text and suggest that human AML leukemogenesis by MLL-fusion oncogenes is likely also influenced by cell cycle rate of the initiating cells.

4. The authors proposed that the rapidly-cycling more immature myeloid progenitors are the cell-of-origin of MLL-AF9-mediated transformation, however no evidence was provided to define the molecular and functional characteristics of these more immature progenitors. Thus scRNA-seq based studies may help better define the cell populations with high transformation potential.

Thank you for this thoughtful suggestion. We have examined the highly proliferative cell state in more detail by scRNA-seq and other assays. The main findings are summarized below.

- 1) We have performed scRNA-seq of freshly isolated GMPs, and defined a subset of GMPs by their low expression levels of *Ly6c2* and *Csf1r*, as the protein product of these genes define the “multilineage” GMPs (Yanez et al., 2015). These cells also express low levels of *Irf8* and *Cebpe* (Fig. 8f, Extended Data Fig. 10b-c), two genes previously reported to mark monocytic and granulocytic commitment. This gene expression pattern suggest that they likely represent the metastable GMPs by (Olsson et al., 2016).
- 2) We prospectively isolated this subset based on surface protein markers, Ly6C and CD115 (new Extended Data Fig. 10a). The Ly6C- CD115^{lo} cells express higher levels of *Cdk4/6* (new Fig. 8e) and undergo cell cycle acceleration during the second day of culture. They display similar cell cycle rate heterogeneity as the other two committed progenitors, the Ly6C+CD115^{lo} granulocyte progenitors and the Ly6C+CD115^{hi} monocyte progenitors (new Fig. 8a-d). Within this subset, transformation efficiency remains correlated with the cell’s cycling rate (new Fig. 8h).

5. The illustration for Fig. 2a,b should be improved, as currently presented, it is very difficult to understand and compare the differences between various conditions (0-24h vs 24-48h, -Dox vs +Dox, etc).

We apologize for the confusion and have revised the figure and text for better clarity.

The plots shown in Fig. 2a-b are similar to a flow cytometry plot, where the pseudo-color denotes population density. Higher values (redder in color) indicate more GMP lineages in a specific region of the plot. The revised figures have contour lines to better indicate this feature. We have also revised the figure legend to the following:

“Cell cycle rates of individual iMLL-AF9 GMP lineages as determined by the number of divisions during the first 24-hours (x-axis) or the second 24-hours (y-axis) of culture. Dox was added at 0 hour, when the presence of single GMP was confirmed. Each dot denotes an individual GMP lineage. (a) -Dox: n=268; (b) +Dox: n=378. The number of divisions during the first 24-hour = $\text{Log}_2(\text{cell number at T24h} / \text{cell number at T0h})$; the number of divisions during the second 24-hour = $\text{Log}_2(\text{cell number at T48h} / \text{cell number at T24h})$. Pseudo color denotes GMP lineage density. Dotted diagonal line denotes $y=x$, indicating comparable cell cycle rates during the first and second 24-hours of culture. In the presence of Dox, significant number of GMP lineages underwent cell cycle acceleration, shown as the dense population above the diagonal line.”

6. The authors used the top enriched GO terms for correlative analysis of changes in cell cycle, proliferation, or differentiation-related genes. GO type of analysis can often be affected by the number of differentially expressed genes, thus it is important to show expression of individual representative genes besides GO terms.

The expression of a panel of representative genes has been added in Extended Data Fig. 8d.

7. Some corroborative evidence is needed for the cell cycle analysis of palbociclib treatment to assess the extent to which the cell cycle was altered.

We have corroborated cell cycle inhibition by palbociclib using a number of complimentary approaches, and the results are shown in Extended Data Fig. 7 and Extended Data Fig. 8.

- 1) We determined the dose and duration of palbociclib treatment on proliferation by scoring cell numbers (Extended Data Fig. 7a), which indicated that 500nM palbociclib treatment reduced total cell numbers. Furthermore, the palbociclib effect is reversible, as proliferation (scored by cell numbers) recovered following palbociclib washout (Extended Data Fig. 7c).
- 2) The reduced cell cycle is reflected by decreased percentage of cells in S/G2/M phase as indicated by cells displaying $>2N$ DNA content (Extended Data Fig. 7b).
- 3) Palbociclib treatment reduced the subset of GMPs that could divide 3 or more times within the first 24-hours (Extended Data Fig. 7d).
- 4) On gene expression level, palbociclib treatment led to significant downregulation of cell cycle genes by gene set enrichment analysis (GSEA) (Extended Data Fig. 8a) without inducing apoptosis (Extended Data Fig. 8b).

8. Statistical analysis is needed for several figure panels (i.e. Fig. 2, Fig. 3c-f, and Fig. 5b) since many of the observed effects was modest.

We have performed statistical tests and added the details in corresponding figures/legends to the suggested figures and others where applicable.

Fig. 2c,d: Unpaired t-test for each division rate between +Dox and -Dox conditions. All $p > 0.1$, except for $p = 0.0036$ for 2 divisions in Fig. 2d. Therefore, the cell cycle rate of GMPs +/- Dox remained identical during 0-24hr culture, but become increased by Dox during 24-48hr culture.

Fig. 2e: Linear regression and F-test to evaluate whether the slopes of linear regression is significantly non-zero. For +Dox condition, $y = 28.06x - 0.4914$, $p < 0.0001$; for -Dox condition,

$y=0.08382x+1.778$, $p=0.9466$. Therefore, cell cycle rate positively correlates with transformation efficiency in +Dox, but not in -Dox condition.

Similar tests were done in Fig. 2g,h.

Fig 3c (new Fig. 3a): Unpaired t-test between indicated samples. All $p<0.01$.

Fig. 3e (new Fig. 3d): We performed nonparametric Kolmogorov-Smirnov test between the TPMs of Dox-up-regulated DEGs and Dox-down-regulated DEGs. $p<0.0001$. Therefore, these genes are expressed at different levels in fresh GMPs.

Fig. 3f (new Fig. 3e): Unpaired t-test between indicated samples. $p<0.01$ for all pair-wise comparisons between fresh and cultured GMPs. Note the fold-change between fresh and +Dox GMPs always appear milder than the fold-change between fresh and -Dox GMPs.

Fig. 5b (new Fig. 6j): Unpaired t-test were used to compare cell numbers. On day 2, $p=0.1489$ between +Dox+Pal and +Dox-Pal; $p=0.1331$ between -Dox+Pal and -Dox-Pal. These results confirmed that the palbociclib condition we used is very mild and did not cause significant cell number changes at this early time point.

References

- Bernt, K.M., Zhu, N., Sinha, A.U., Vempati, S., Faber, J., Krivtsov, A.V., Feng, Z.H., Punt, N., Daigle, A., Bullinger, L., *et al.* (2011). MLL-Rearranged Leukemia Is Dependent on Aberrant H3K79 Methylation by DOT1L. *Cancer Cell* 20, 66-78.
- Buenostro, J.D., Corces, M.R., Lareau, C.A., Wu, B., Schep, A.N., Aryee, M.J., Majeti, R., Chang, H.Y., and Greenleaf, W.J. (2018). Integrated Single-Cell Analysis Maps the Continuous Regulatory Landscape of Human Hematopoietic Differentiation. *Cell* 173, 1535-1548 e1516.
- Farrar, J.E., Schuback, H.L., Ries, R.E., Wai, D., Hampton, O.A., Trevino, L.R., Alonzo, T.A., Guidry Auvil, J.M., Davidsen, T.M., Gesuwan, P., *et al.* (2016). Genomic Profiling of Pediatric Acute Myeloid Leukemia Reveals a Changing Mutational Landscape from Disease Diagnosis to Relapse. *Cancer Res* 76, 2197-2205.
- Guo, S., Bai, H., Megyola, C.M., Halene, S., Krause, D.S., Scadden, D.T., and Lu, J. (2012). Complex oncogene dependence in microRNA-125a-induced myeloproliferative neoplasms. *Proc Natl Acad Sci U S A* 109, 16636-16641.
- Guo, S., Zi, X., Schulz, V.P., Cheng, J., Zhong, M., Koochaki, S.H., Megyola, C.M., Pan, X., Heydari, K., Weissman, S.M., *et al.* (2014). Nonstochastic reprogramming from a privileged somatic cell state. *Cell* 156, 649-662.
- Herault, A., Binnewies, M., Leong, S., Calero-Nieto, F.J., Zhang, S.Y., Kang, Y.A., Wang, X., Pietras, E.M., Chu, S.H., Barry-Holson, K., *et al.* (2017). Myeloid progenitor cluster formation drives emergency and leukaemic myelopoiesis. *Nature* 544, 53-58.
- Olsson, A., Venkatasubramanian, M., Chaudhri, V.K., Aronow, B.J., Salomonis, N., Singh, H., and Grimes, H.L. (2016). Single-cell analysis of mixed-lineage states leading to a binary cell fate choice. *Nature* 537, 698-+.
- Stavropoulou, V., Almosaillekh, M., Royo, H., Spetz, J.-F., Juge, S., Brault, L., Kopp, P., Iacovino, M., Kyba, M., Tzankov, A., *et al.* (2018). A Novel Inducible Mouse Model of MLL-ENL-driven Mixed-lineage Acute Leukemia. 2, e51.
- Yanez, A., Ng, M.Y., Hassanzadeh-Kiabi, N., and Goodridge, H.S. (2015). IRF8 acts in lineage-committed rather than oligopotent progenitors to control neutrophil vs monocyte production. *Blood* 125, 1452-1459.

REVIEWERS' COMMENTS:

Reviewer #1 (Remarks to the Author):

The authors have addressed all my points

Reviewer #2 (Remarks to the Author):

The authors did a very good job revising their manuscript and strengthening their conclusions. I have no other comments.

Reviewer #3 (Remarks to the Author):

The authors are to be commended for the large amount of new experiments and analyses that they have performed to address my original comments and the concerns of the other reviewers. The new scRNA-seq analysis comparing freshly isolated and cultured GMPs with or without Dox-induced MLL-AF9 expression (Figs. 4, S5 and S10) provided new insight into the regulation of cell cycle states and the transforming potential by MLL-AF9 in different GMP subsets. The results support the main conclusions that the cell cycle heterogeneity in GMPs is the major determinant of the probability of transformation by the MLL-AF9 fusion oncogene. The authors also provide new evidences such as the analysis of cell cycle rates in different GMP subsets, the analysis of MLL-AF9-mediated transformation after modulations of cell cycle rates, and the analysis of CCND1 expression in different human GMP subsets and the correlation with survival in the TARGET-AML cohort, etc.

Overall the new data and textual revisions significantly enhanced the strength of the conclusions and addressed my original questions. This study provides an important example that the difference in cell cycle rate may be the major determining factor in the transformation potential by certain oncogenic drivers such as MLL-AF9. This reviewer only had a few minor points for the revised manuscript as following.

- 1) Figs. 7c and 8a,b,h, it would be helpful to show mean +/- s.d. or s.e.m. of the data as other similar figures (e.g. Fig. 2c,h).
- 2) The authors should consider consolidating some of the main figures (currently 9 main figures) and/or removing some of the less essential items to the supplemental figures.
- 3) Line 146, typo 'corelate' should be 'correlate'.

REVIEWERS' COMMENTS:

Our answers are in this color.

Reviewer #1 (Remarks to the Author):

The authors have addressed all my points

Reviewer #2 (Remarks to the Author):

The authors did a very good job revising their manuscript and strengthening their conclusions. I have no other comments.

Reviewer #3 (Remarks to the Author):

The authors are to be commended for the large amount of new experiments and analyses that they have performed to address my original comments and the concerns of the other reviewers. The new scRNA-seq analysis comparing freshly isolated and cultured GMPs with or without Dox-induced MLL-AF9 expression (Figs. 4, S5 and S10) provided new insight into the regulation of cell cycle states and the transforming potential by MLL-AF9 in different GMP subsets. The results support the main conclusions that the cell cycle heterogeneity in GMPs is the major determinant of the probability of transformation by the MLL-AF9 fusion oncogene. The authors also provide new evidences such as the analysis of cell cycle rates in different GMP subsets, the analysis of MLL-AF9-mediated transformation after modulations of cell cycle rates, and the analysis of CCND1 expression in different human GMP subsets and the correlation with survival in the TARGET-AML cohort, etc.

Overall the new data and textual revisions significantly enhanced the strength of the conclusions and addressed my original questions. This study provides an important example that the difference in cell cycle rate may be the major determining factor in the transformation potential by certain oncogenic drivers such as MLL-AF9. This reviewer only had a few minor points for the revised manuscript as following.

1) Figs. 7c and 8a,b,h, it would be helpful to show mean +/- s.d. or s.e.m. of the data as other similar figures (e.g. Fig. 2c,h).

We have added error bars to show mean +/-S.D in Figs. 7b, 7c, 8a, 8b, 8d, 8h, and Supplementary Fig. 10d.

2) The authors should consider consolidating some of the main figures (currently 9 main figures) and/or removing some of the less essential items to the supplemental figures.

We thank the reviewer for this suggestion and have revised accordingly. Specifically, the original Fig. 5c and 5e are presented as new Supplementary Fig. 6c and 6f; the original Fig. 6i, j, l, m are presented as new Supplementary Fig. 8a, b, f, g.

3) Line 146, typo 'corelate' should be 'correlate'.

Corrected.